# Loopholing Discrete Diffusion: Deterministic Bypass of the Sampling Wall

**Mingyu Jo[1*], Jaesik Yoon[1,4], Justin Deschenaux[2], Caglar Gulcehre[2,3], Sungjin Ahn[1,5*]**
[1]KAIST, [2]EPFL, [3]Microsoft, [4]SAP, [5]NYU

## Abstract

Discrete diffusion models offer a promising alternative to autoregressive generation through parallel decoding, but they suffer from a sampling wall: once categorical sampling occurs, rich distributional information collapses into one-hot vectors and cannot be propagated across steps, forcing subsequent steps to operate with limited information. To mitigate this problem, we introduce Loopholing, a novel and simple mechanism that preserves this information via a deterministic latent pathway, leading to Loopholing Discrete Diffusion Models (LDDMs). Trained efficiently with a self-conditioning strategy that avoids unrolling the full denoising trajectory, LDDMs achieve substantial gains—reducing generative perplexity by up to 61% over prior baselines, thereby closing (and in some cases surpassing) the gap with autoregressive models, and producing more coherent text. Applied to reasoning tasks, LDDMs also improve performance on arithmetic benchmarks such as Countdown and Game of 24. These results also indicate that loopholing mitigates idle steps and oscillations, providing a general and effective path toward high-quality non-autoregressive text generation.

## 1 Introduction

Discrete diffusion models have recently emerged as a promising alternative to autoregressive models for tasks such as text generation (Austin et al., 2021; Campbell et al., 2022; Lou et al., 2023; Sahoo et al., 2024; Schiff et al., 2024; Zhao et al., 2024; Ou et al., 2024; Gat et al., 2024; Wang et al., 2025a). Unlike autoregressive models, which generate tokens sequentially from left to right, discrete diffusion generates entire sequences in parallel through iterative refinement across multiple denoising steps. This parallel generation not only enables substantial speedups for long-horizon sequence generation but also allows the model to leverage global sequence context rather than being restricted to left-context only.

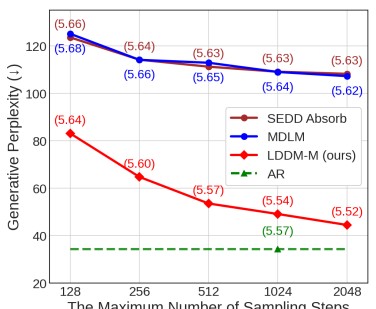

**Figure 1:** Unconditional gen PPL measured with GPT-2 Large (sentence entropy in parentheses).

However, despite these advantages, empirical studies show that discrete diffusion models still lag behind autoregressive models in generation quality (Gat et al., 2024; Zheng et al., 2024), indicating a substantial room for improvement. For example, recent studies have highlighted specific issues in discrete diffusion models, such as *idle steps* (Chao et al., 2025) and *temporal oscillation* (Wang et al., 2025b).

As part of such effort to improve discrete diffusion models, in this work we first focus on a fundamental phenomenon that may underlie these inefficiencies, which we term the *sampling wall*. We define the sampling wall as a form of information collapse, in which rich categorical distributions—capturing plausible token candidates and their relative likelihoods—are reduced to one-hot vectors after sampling. We call this a "wall" because, in standard discrete diffusion models, once sampling occurs, the original distributional information is lost and cannot be propagated to subsequent steps. Motivated by this observation, our central hypothesis is that explicitly propagating distributional information beyond the sampling wall across denoising steps can alleviate key limitations of discrete diffusion models.

---

*Correspondence to Mingyu Jo (mingyu.jo@kaist.ac.kr) and Sungjin Ahn (sungjin.ahn@kaist.ac.kr).

In this paper, to realize this idea, we propose a simple and novel mechanism, termed *Loopholing*, together with a corresponding family of models, *Loopholing Discrete Diffusion Models (LDDMs)*. Our key idea in the *Loopholing* mechanism is to introduce a direct, deterministic pathway that transfers the rich contextual latent state to the subsequent step. This pathway complements the existing stochastic path; thus, in *Loopholing*, each denoising step produces two outputs: a stochastic one-hot vector and a deterministic continuous vector. While this design introduces a recurrent dependency across the denoising trajectory, which would require full unrolling for training, we make training feasible at randomly sampled time steps without unrolling by introducing a self-conditioning approach (Chen et al., 2022; Jabri et al., 2022) tailored for *Loopholing*.

The main contributions of the paper are as follows. (i) **Identifying the Sampling Wall Problem**: We identify the *sampling wall* problem as a fundamental characteristic that may underlie various inefficiencies in the standard discrete diffusion models. (ii) **Introducing Loopholing**: We propose the Loopholing mechanism, and Loopholing Discrete Diffusion Models (LDDMs). (iii) **Strong Empirical Results**: We demonstrate the effectiveness of LDDMs through various experiments. On the OpenWebText dataset (Gokaslan & Cohen, 2019), our model improves the MDLM (Sahoo et al., 2024) test perplexity from 23.82 to 21.90. In terms of Generation Perplexity (Gen PPL), as shown in Fig. 1, our method achieves substantial gains, reducing Gen PPL by 55% relative to MDLM and by 61% relative to UDLM. Against autoregressive models, the gap shrinks from $3.17\times$ higher Gen PPL with MDLM to only $1.43\times$ with our method. Remarkably, when applied to UDLM, our approach not only closes the gap but even surpasses the autoregressive baseline, whereas the standard UDLM lags behind by $2.15\times$. In reasoning, loopholing mechanism boosts the accuracy on Countdown4 (Gandhi et al., 2024) from 45% to 56.3% over the MGDM baseline (Ye et al., 2024).

## 2 PRELIMINARIES

**Discrete Diffusion Models.** Diffusion models (Sohl-Dickstein et al., 2015; Ho et al., 2020) are probabilistic generative models defined by a fixed forward process that progressively corrupts data with noise, and a learned reverse process trained to recover the original sample $\mathbf{x}$. For discrete data, this framework is adapted by representing a categorical data sample as a one-hot vector $\mathbf{x} \in \mathcal{V}$, where $\mathcal{V} = \{v \in \{0,1\}^K : \sum_k v_k = 1\}$ is the set of all one-hot vectors over a vocabulary of size $K$. For clarity, we describe the formulation in the simplest case of a single categorical variable.

The forward process corrupts the initial data sample $\mathbf{x}$ over a continuous time variable $t \in [0,1]$, producing a sequence of progressively noisier latent variables $\mathbf{z}_t$. A common approach for this is to use an interpolating discrete diffusion model (Sahoo et al., 2024; Schiff et al., 2024; Shi et al., 2024; von Rütte et al., 2025), where the marginal distribution of the latent state $\mathbf{z}_t$, conditioned on the original data $\mathbf{x}$, is formulated as a categorical distribution that interpolates between the data $\mathbf{x}$ and a fixed prior distribution $\boldsymbol{\pi}$:

$$q(\mathbf{z}_t|\mathbf{x}) = \mathrm{Cat}(\mathbf{z}_t; \alpha_t \mathbf{x} + (1 - \alpha_t)\boldsymbol{\pi}), \tag{1}$$

where $\mathrm{Cat}(\cdot; \mathbf{p})$ denotes a categorical distribution parameterized by probability simplex $\mathbf{p}$, and $\alpha_t \in [0,1]$ is a monotonically decreasing noise schedule with $\alpha_0 \approx 1$ and $\alpha_1 \approx 0$.

**Masked Diffusion Models (MDMs)** are a subclass of discrete diffusion models in which the prior $\boldsymbol{\pi}$ is set to $\mathbf{m}$, the one-hot vector corresponding to a special $[\texttt{MASK}]$ token (Sahoo et al., 2024; Nie et al., 2025; Kim et al., 2025). Under this formulation, the forward process can be interpreted as the gradual masking of tokens over time.

In MDMs, the reverse process generates data by progressively denoising a sample, beginning from the prior distribution in which all tokens are masked. Formally, it seeks to approximate the true reverse posterior $q(\mathbf{z}_s|\mathbf{z}_t, \mathbf{x})$ for any $0 \le s < t \le 1$. In practice, this distribution is parameterized using a neural network $\mathbf{x}_\theta(\mathbf{z}_t, t)$ trained to predict the original data $\mathbf{x}$. In the following, we use $\mathbf{x}_{\theta,t}$ as a shorthand for $\mathbf{x}_\theta(\mathbf{z}_t, t)$, whenever this does not cause confusion. The resulting approximation of the posterior takes the form:

$$q(\mathbf{z}_s|\mathbf{z}_t, \mathbf{x}_{\theta,t}) = \begin{cases} \delta_{\mathbf{z}_t}(\mathbf{z}_s), & \mathbf{z}_t \neq \mathbf{m}, \\ \mathrm{Cat}\left(\mathbf{z}_s; \frac{(1-\alpha_s)\mathbf{m} + (\alpha_s - \alpha_t)\mathbf{x}_{\theta,t}}{1 - \alpha_t}\right), & \mathbf{z}_t = \mathbf{m}, \end{cases} \tag{2}$$

where $\delta_{\mathbf{x}_t}$ denotes the Dirac delta function. This ensures unmasked tokens are preserved, while masked tokens are resampled according to the model's predictions and the given schedule.

For training, MDMs optimize a simplified Negative Evidence Lower Bound (NELBO). In continuous time, this specific formulation provides a tighter objective than its discrete-time counterparts (Austin et al., 2021; Kingma et al., 2021). In practice, the objective reduces to a weighted cross-entropy loss (Sahoo et al., 2024; Shi et al., 2024):

$$\mathcal{L}_{\text{NELBO}} = \mathbb{E}_{t \sim \mathcal{U}[0,1], \mathbf{z}_t \sim q(\mathbf{z}_t|\mathbf{x})} \left[ \mathbb{I}[\mathbf{z}_t = \mathbf{m}] \frac{\alpha'_t}{1 - \alpha_t} \log \langle \mathbf{x}_\theta(\mathbf{z}_t, t), \mathbf{x} \rangle \right] , \tag{3}$$

where $\mathcal{U}[0, 1]$ denotes the uniform distribution, $\alpha'_t$ is the time derivative of $\alpha_t$, $\langle \cdot, \cdot \rangle$ represents the dot product, and $\mathbb{I}[\cdot]$ is the indicator function, returning 1 if the condition holds and 0 otherwise. This objective encourages the model to accurately reconstruct the original tokens at masked positions.

Besides MDMs, there are also Uniform Diffusion Models (UDMs), which use a uniform distribution over the entire vocabulary as the prior $\boldsymbol{\pi}$ (Austin et al., 2021; Schiff et al., 2024; Sahoo et al., 2025). Further details on UDMs are provided in Appendix B.

## 3 LOOPHOLING DISCRETE DIFFUSION MODELS

To begin with, we first discuss a characteristic of discrete diffusion models that motivated the design of the *Loopholing* mechanism. We refer to it as the *sampling wall* problem.

**The sampling wall problem** represents a form of information collapse, where rich categorical distributional representations are reduced to one-hot vectors. Specifically, while $\mathbf{x}_{\theta,t} = \mathbf{x}_\theta(\mathbf{z}_t, t)$ encodes far richer information about plausible token candidates and their relative likelihoods, this information is discarded once a single category $\mathbf{z}_s$ is drawn, leaving only the one-hot representation to be propagated forward. For example, consider two cases: $\mathbf{x}^a_{\theta,t} = [0.49, 0.51]$ and $\mathbf{x}^b_{\theta,t} = [0.20, 0.80]$. Despite reflecting very different situations, the denoising process cannot distinguish between the two if the second category is sampled in both cases, entirely discarding the predictive distribution at that position. Without propagating this distributional information, a process based solely on sampling can become redundant (Chao et al., 2025) and prone to excessive oscillations (Wang et al., 2025b).

To address the sampling wall problem in discrete diffusion models, we propose a novel mechanism, termed *Loopholing*, together with a corresponding family of models, *Loopholing Discrete Diffusion Models (LDDMs)*. The goal of LDDMs is to operationalize the following central hypothesis in both the architecture and the training procedure of discrete diffusion models:

> **Main Hypothesis**: Enriching the denoising process by propagating detailed contextual information available prior to sampling discrete tokens—such as the categorical distribution parameter $\mathbf{x}_{\theta,t}$—can alleviate the aforementioned issues and leads to improved performance.

Our key idea for implementing the above hypothesis in the loopholing mechanism is to introduce, alongside the standard sampling pathway, a direct deterministic pathway that carries rich distributional context information obtained *before sampling* across denoising steps.

In the following, we present Loopholing Discrete Diffusion Models (LDDMs) in two parts. First, we describe the generation process of LDDMs, detailing how distributional context is obtained and propagated throughout denoising. Second, we explain how LDDMs can be trained efficiently through a self-conditioning approach. Notably, both the generation and training mechanisms of Loopholing are straightforward to implement, requiring only minor modifications to the standard discrete diffusion framework.

### 3.1 GENERATION WITH LOOPHOLING

In standard discrete diffusion models, as shown in Fig. 2(a), the modeling of the denoising process operates on a sequence of tokens $\mathbf{z}_t^{(1:L)} = [\mathbf{z}_t^{(1)}, \dots, \mathbf{z}_t^{(L)}]$. First, for each position $\ell$ in the sequence, it converts the one-hot input token $\mathbf{z}_t^{(\ell)}$ into an embedding $E_\theta(\mathbf{z}_t^{(\ell)})$. The embeddings of the sequence are passed through a backbone network such as a Transformer (Vaswani et al., 2017) layer to mix the sequence embeddings and produce a latent embedding $\mathbf{h}_s^{(\ell)}$. Finally, this latent embedding is fed into a projection layer to predict the distribution of the clean observation at that position $\mathbf{x}_\theta^{(\ell)}(\mathbf{z}_t^{(1:L)}, t)$.

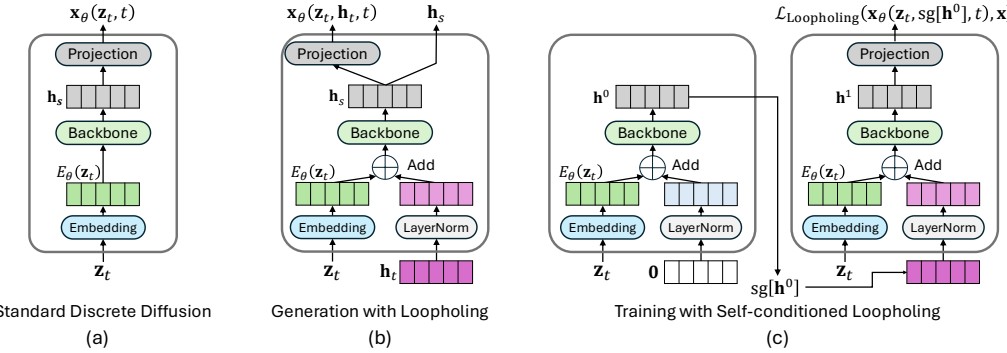

**Figure 2:** Architectural comparison of standard discrete diffusion models and the Loopholing Discrete Diffusion Models (LDDMs). **(a)** The standard architecture of discrete diffusion. **(b)** During inference, LDDMs propagate the continuous latent representation $\mathbf{h}_s$ to the subsequent step, creating a deterministic pathway that preserves rich contextual information. **(c)** During training, LDDMs employ a self-conditioning strategy: a first pass generates a pseudo-context $\mathbf{h}^0$, which is then used to condition the second pass.

Crucially, while the backbone aggregates global context across the sequence, the sampling step applies the posterior in Eqn. 2 independently to each token position. Therefore, for notational clarity, we describe the subsequent formulation in the simplest case of a single categorical variable $\mathbf{z}_t$, omitting the positional index $\ell$.

We observe that rich contextual information is captured in both $\mathbf{h}_s$ and $\mathbf{x}_\theta(\mathbf{z}_t, t)$ during this process. This information is rich because it accounts for complex relational interactions among tokens and is represented as a high-dimensional continuous vector rather than a one-hot encoding. However, once the output representation $\mathbf{z}_s$ is sampled, this rich information collapses into a one-hot vector. Therefore, the next denoising step, which takes only $\mathbf{z}_s$ as input, cannot exploit this information or build upon it, and must instead reconstruct much of it again from the limited one-hot representation.

From this observation, our key idea in the loopholing mechanism is to introduce a direct, deterministic pathway that transfers the rich contextual latent state $\mathbf{h}_s$ to the subsequent step. This pathway complements the existing stochastic path; thus, in loopholing, each denoising step produces two outputs: a stochastic one-hot vector and a deterministic continuous vector:

$$(\mathbf{x}_\theta(\mathbf{z}_t, \mathbf{h}_t, t), \mathbf{h}_s) = f_{\text{Loopholing}}(\mathbf{z}_t, \mathbf{h}_t, t). \tag{4}$$

Formally, the denoising process with loopholing as shown in Fig. 2(b) is described as follows. Let $\mathbf{z}_t$ be the one-hot vector for a token at step $t$. We initialize a latent state $\mathbf{h}_1 = \mathbf{0}$. At each denoising step $t \to s$, the model performs the following computations:

$$\mathbf{e}_t = E_\theta(\mathbf{z}_t) + \text{LN}(\mathbf{h}_t), \quad \mathbf{h}_s = f_\theta(\mathbf{e}_t, t), \quad \mathbf{x}_\theta(\mathbf{z}_t, \mathbf{h}_t, t) = \text{softmax}(g_\theta(\mathbf{h}_s)), \tag{5}$$

where $E_\theta$ is the token embedding function, $f_\theta$ is the backbone network, and $g_\theta$ is the output projection layer. The previous latent embedding $\mathbf{h}_t$ combines with the current token embedding $E_\theta(\mathbf{z}_t)$ via Layer Normalization (LN) (Ba et al., 2016), creating a deterministic contextual latent path. This prediction $\mathbf{x}_\theta(\mathbf{z}_t, \mathbf{h}_t, t)$ is used to parameterize the reverse posterior (Eqn. 2) and sample $\mathbf{z}_s$.

Note that while it is also possible to pass the model's prediction $\mathbf{x}_\theta(\mathbf{z}_t, \mathbf{h}_t, t)$, to the subsequent step, this distribution over the vocabulary typically has much higher dimensionality than $\mathbf{h}_t$ in important applications such as language modeling. For this reason, we pass $\mathbf{h}_t$ in our default architecture.

## 3.2 TRAINING WITH SELF-CONDITIONED LOOPHOLING

A key advantage of diffusion models is that training does not require time-consuming temporal unrolling. Instead, training can be performed on randomly sampled time steps, as $q(\mathbf{z}_t|\mathbf{x})$ can be directly constructed for any arbitrary time step. However, the generation with loopholing requires propagating $\mathbf{h}_t$ across denoising steps, which introduces a recurrent dependency. Maintaining this training efficiency within loopholing is therefore a significant challenge.

To address this, we introduce a self-conditioning approach (Chen et al., 2022; Jabri et al., 2022) to avoid unrolling the full generation path during training. The core idea, illustrated in Fig. 2(c), is to

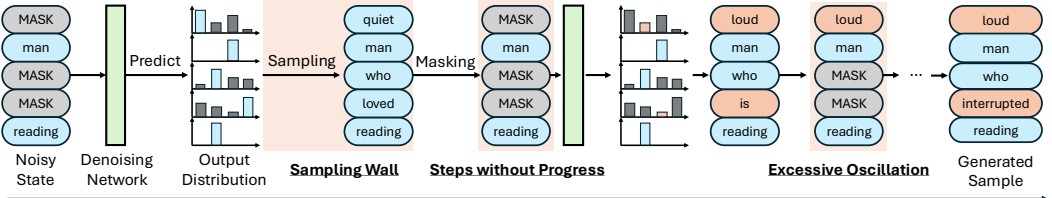

**Figure 3:** Illustration of the *sampling wall* in Masked Diffusion Models (MDMs), which induces two distinct failure modes. (1) **Steps without Progress**: Fixing on a single token can cause the input sequence to remain static across multiple denoising steps, leading to significant computational inefficiency. (2) **Excessive Oscillations**: Sampling a low-probability token (e.g., "loud") can trigger excessive oscillations in subsequent steps.

simulate the context propagation process using two forward passes: for a given noisy input $\mathbf{z}_t$, the model first computes a pseudo-context $\mathbf{h}^0$ and then uses it in a second, context-conditioned pass as input to make the final prediction. Specifically, the process for each training step is as follows:

1. **First Pass (Pseudo-Context Generation):** We perform a loopholing denoising, but by setting the input context state to zero vector. This yields a pseudo-context $\mathbf{h}^0$ and an initial prediction $\mathbf{x}_\theta^0(\mathbf{z}_t, \mathbf{0}, t)$ directly from the loopholing function:

$$(\mathbf{x}_\theta^0(\mathbf{z}_t, \mathbf{0}, t), \mathbf{h}^0) = f_{\text{Loopholing}}(\mathbf{z}_t, \mathbf{h}_t = \mathbf{0}, t) . \tag{6}$$

2. **Second Pass (Context-Conditioned Prediction):** The second pass takes the pseudo-latent embedding $\mathbf{h}^0$ from the first pass as if it is from the previous step during the generation process:

$$(\mathbf{x}_\theta^1(\mathbf{z}_t, \mathbf{h}_t, t), \mathbf{h}^1) = f_{\text{Loopholing}}(\mathbf{z}_t, \mathbf{h}_t = \text{sg}[\mathbf{h}^0], t) . \tag{7}$$

The stop-gradient operator, $\text{sg}[\cdot]$, ensures that gradients flow only through the second forward pass. This allows the model to learn how to consume its own representations as context without the prohibitive cost of backpropagating through time.

This training objective can then be expressed as:

$$\mathcal{L}_{\text{Loopholing}} = \mathbb{E}_{t \sim \mathcal{U}[0,1], \mathbf{z}_t \sim q(\mathbf{z}_t|\mathbf{x})} \left[ \mathbb{I}[\mathbf{z}_t = \mathbf{m}] \frac{\alpha_t'}{1 - \alpha_t} \left( \log \langle \mathbf{x}_\theta^1(\mathbf{z}_t, \text{sg}[\mathbf{h}^0], t), \mathbf{x} \rangle \right) \right] , \tag{8}$$

where the expectation is taken over a uniformly sampled time $t \sim \mathcal{U}[0, 1]$ and the corresponding noised input $\mathbf{z}_t \sim q(\mathbf{z}_t|\mathbf{x})$. Furthermore, following previous work (Jabri et al., 2022), we employ self-conditioning with a probability of $p$. Specifically, the model is trained on the self-conditioning loss with probability $p$, and on the standard discrete diffusion loss (Eqn. 3) otherwise. This approach encourages the contextual latent from the first pass to accurately capture the context of $\mathbf{x}$ while providing useful guidance for the second pass.

## 4 DISCUSSION: WHY LOOPHOLING WORKS

We hypothesize that the sampling wall manifests through two key inefficiencies in discrete diffusion, which are illustrated in Fig. 3.

**Steps without progress:** As shown in a recent study (Chao et al., 2025), many denoising steps in standard discrete diffusion models reproduce the same samples as in previous steps, resulting in idle steps (see Appendix D.8). This is wasteful because it does not make any progress in the sequence. We hypothesize that loopholing provides a way to address this issue by enabling continuous updates to the context latent $\mathbf{h}_t$ even when the sample $\mathbf{z}_t$ remains unchanged. Specifically, the deterministic recurrent state update in the $\mathbf{h}_t$ space (Eqn. 4) ensures that contextual information evolves across steps, so that each iteration contributes to refining the output. As shown in our ablation study (Section 6.3), LDDMs indeed exhibit higher Temporal KL divergence during the early denoising phase, confirming that each step makes more meaningful progress along the denoising trajectory. This improved efficiency allows high-quality samples to be generated with fewer denoising steps.

**Table 1:** Comparison of test perplexities (↓) of models trained for 1 million steps on the One Billion Word (LM1B) and OpenWebText (OWT) datasets. † denotes retrained model.

|  | LM1B | OWT |
|---|---|---|
| *Masked Diffusion* | | |
| SEDD Absorb†(Lou et al., 2023) | $\leq 28.39$ | $\leq 24.01$ |
| MDLM† (Sahoo et al., 2024) | $\leq 27.60$ | $\leq 23.05$ |
| *Uniform Diffusion* | | |
| UDLM† (Schiff et al., 2024) | $\leq 31.11$ | $\leq 25.51$ |
| *Ours (LDDMs)* | | |
| LDDM-M (ours) | $\leq \mathbf{25.95}$ | $\leq \mathbf{21.90}$ |
| LDDM-U (ours) | $\leq 29.21$ | $\leq 23.82$ |

**Excessive Oscillation:** The sampling wall can induce oscillations during denoising. Although standard discrete diffusion models consistently predict the same objective, $\mathbf{x}_{\theta,t}$, at each step, it discards the rich distributional information from the prior step's prediction. This forces the model to predict from scratch, relying only on the current, stochastically sampled sequence (Wang et al., 2025b). We hypothesize that the loopholing mechanism can lead to a more stable generation process. By providing a deterministic information path, this mechanism enables the model to directly maintain contextual information about the target $\mathbf{x}$ throughout the denoising process. Indeed, our ablation (Section 6.3) shows that LDDMs exhibit lower Temporal KL divergence in the later denoising phase and consistently lower Token-Prediction Entropy, indicating more stable generation.

## 5 RELATED WORKS

**Discrete Diffusion for Language Modeling**. The pioneering work (Austin et al., 2021) introduced discrete denoising diffusion using transition matrices based on an absorbing "mask" state and uniform noise. Subsequent models, including Score Entropy Discrete Diffusion (SEDD) (Lou et al., 2023), Mask Diffusion Language Models (MDLM) (Sahoo et al., 2024), Uniform Diffusion Language Models (UDLM) (Schiff et al., 2024) and Duo (Sahoo et al., 2025), have further advanced this paradigm with improved training objectives, producing stronger language modeling performance. However, a shared limitation is the reliance on repeated categorical sampling across the sequence, which can degrade contextual coherence. Our work directly addresses this challenge.

**Self-Conditioning**. Self-conditioning techniques improve consistency across generative steps by reusing previous model outputs or hidden states during training. For instance, Analog Bits (Chen et al., 2022) employs self-conditioning to enhance sampling performance, while Recurrent Interface Networks (RINs) (Jabri et al., 2022) use it to reduce the computational cost of training by avoiding backpropagation throughout the generation trajectory. Inspired by these approaches, our loopholing mechanism integrates a self-conditioning strategy to efficiently train the model to use the propagated latent embedding as internal memory.

## 6 EXPERIMENTS

In this section, we empirically validate the effectiveness of the proposed loopholing mechanism across various models and tasks. We demonstrate that **by accumulating contextual information, our mechanism achieves superior perplexity and generation quality in language modeling, along with higher success rates on reasoning tasks.** To further understand these improvements, we conduct a series of ablation studies that provide a detailed analysis of our method's key components. For reproducibility, we provide full experimental details, including datasets, hyperparameters, training procedures, and evaluation protocols, in Appendix C.

### 6.1 LANGUAGE MODELING

To demonstrate the efficacy of our loopholing mechanism, we first apply it to discrete diffusion language models. Specifically, we integrate our method into the Masked Diffusion Language Mod-

**Table 2:** Zero-shot perplexities (↓) after 1 million training steps on OpenWebText. All reported perplexity values are upper bounds. † denotes retrained model.

|  | PTB | Wikitext | LM1B | Lambada | AG News | Pubmed | Arxiv |
|---|---|---|---|---|---|---|---|
| *Masked Diffusion* | | | | | | | |
| SEDD Absorb† | 97.87 | 38.34 | 74.71 | 50.15 | 76.54 | 45.25 | 39.75 |
| MDLM† | 86.33 | 36.30 | **66.73** | 48.36 | 68.62 | 41.94 | 37.52 |
| *Uniform Diffusion* | | | | | | | |
| UDLM† | 77.28 | 38.48 | 81.41 | 51.68 | 76.81 | 46.18 | 41.19 |
| *Ours (LDDMs)* | | | | | | | |
| LDDM-M (ours) | 85.80 | **33.27** | 69.53 | **44.22** | **62.55** | **39.74** | **34.96** |
| LDDM-U (ours) | **71.52** | 38.89 | 79.60 | 52.34 | 76.81 | 45.05 | 41.02 |

els (MDLM) (Sahoo et al., 2024) and the Uniform Diffusion Language Models (UDLM) (Schiff et al., 2024), creating **LDDM-M** and **LDDM-U**, respectively. We train these models on the One Billion Word (LM1B) (Chelba et al., 2013) and OpenWebText (OWT) (Gokaslan & Cohen, 2019) datasets. Following the training phase, we evaluate their performance based on three key metrics: likelihood, zero-shot likelihood on unseen datasets, and the quality of the generated samples. For a fair comparison, all models were trained under identical settings, using the same datasets and training configurations, following the setup of Sahoo et al. (2024). A detailed description of our configuration is provided in Appendix C.1.

**Likelihood Evaluation**. As shown in Table 1, **our *loopholing* mechanism consistently improves performance across both masked and uniform diffusion frameworks on the LM1B and OWT datasets.** This result indicates that our approach is effective regardless of the underlying diffusion model type. Perplexity was measured by approximating the Negative Evidence Lower Bound (NELBO; Eqn. 3), with further details on the evaluation protocol available in Appendix C.1.2.

**Zero-Shot Likelihood Evaluation**. To assess the generalization performance of our models, we evaluated the models trained on the OWT dataset across a diverse set of unseen datasets. A detailed description of these evaluated datasets is provided in Appendix C.1.3.

As shown in Table 2, **LDDM-M, our loopholing-enhanced MDLM, consistently outperforms the baseline MDLM on all evaluated unseen datasets except for LM1B.** In contrast, LDDM-U exhibits only marginal improvements over UDLM, with the exception of the PTB dataset. We hypothesize that this performance disparity stems from the fundamental differences in how perplexity is calculated for uniform diffusion framework. Uniform diffusion calculates perplexity over all tokens, making the metric highly sensitive to domain shift between the training and test distributions rather than the effectiveness of the loopholing mechanism.

**Generation Quality Evaluation**. The loopholing mechanism is designed to address the sampling wall issue, a property that is difficult to verify solely through likelihood evaluation. To directly assess its impact on generation quality, we therefore employ two alternative metrics. First, we measure the perplexity of unconditionally generated samples using a pretrained GPT-2 Large model (Radford et al., 2019), which we refer to as generative perplexity (Gen PPL). Second, we utilize GPT-4.1 to score the consistency and naturalness of the samples on a 0-to-10 scale, following the G-eval framework Liu et al. (2023). Specific experimental details are provided in the Appendix C.1.4.

As depicted in Figs. 1 and 4(a), applying loopholing yields substantial improvements in generative perplexity. For instance, at 1024 sampling steps, **LDDM-M achieves a Gen PPL of 49.13, more than halving MDLM's 108.94**. Similarly, **LDDM-U (28.76) shows about 2.5x improvement over UDLM (73.95)**. Critically, this performance gap is not limited to a single step count; unlike their baselines which show signs of saturation, both LDDM-M and LDDM-U exhibit a consistent downward trend in perplexity as the number of sampling steps increases. This demonstrates that loopholing enables meaningful, continuous refinement at each step, directly mitigating the steps without progress issue. Notably, as shown in Fig. 4(a), **LDDM-U also surpasses the strong auto-regressive baseline after 512 steps**. Furthermore, these quality gains are achieved without sacrificing diversity, as evidenced by the stable sentence entropy. This suggests that loopholing improves

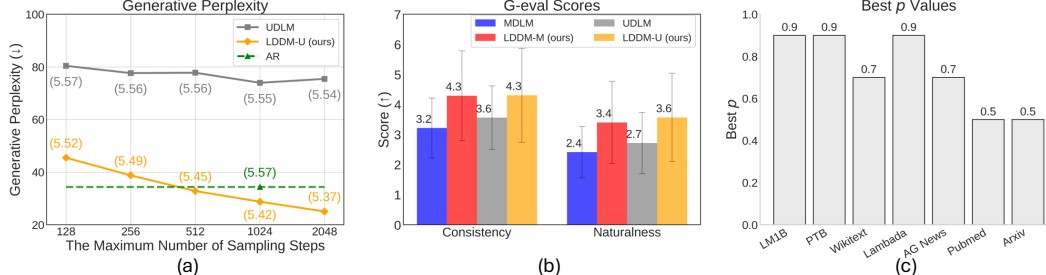

**Figure 4:** **(a)** Unconditional generative perplexity measured using GPT-2 Large, with values in parentheses indicating the sentence entropy of the generated samples. **(b)** Evaluation of generation quality for consistency and naturalness using G-eval framework, rated by GPT-4.1 on a 1-10 scale. **(c)** The optimal value of the self-conditioning rate ($p$) that yields the lowest zero-shot perplexity for each dataset.

generation quality not by collapsing the output distribution, but by guiding the sampling process more effectively within a rich and diverse token space.

The G-eval results in Fig. 4(b) further substantiate these findings on human-aligned metrics. The marked improvement in **consistency** suggests that by maintaining a richer contextual representation throughout the whole generated sequence, loopholing helps the model produce more coherent and logically connected text. Similarly, the higher **naturalness** scores indicate that by propagating the rich contextual latent, the model generates more fluid and human-like sentence structures. Together, these automated and human-aligned evaluations confirm that loopholing provides a robust solution to enhance generation quality in discrete diffusion models. Additional evaluation results on downstream tasks are provided in Appendix D.2.

## 6.2 REASONING TASK

To evaluate the effectiveness of loopholing on reasoning tasks, we integrate it into the Multi-Granularity Diffusion Model (MGDM), a masked diffusion framework designed for reasoning (Ye et al., 2024), resulting in the model we refer to as **LDDM-G**. We evaluate its performance on two arithmetic reasoning tasks that require high logical precision: Countdown (Gandhi et al., 2024) and Game of 24 (Yao et al., 2023). The objective in these tasks is to generate a valid arithmetic formula that yields a target number using a given set of digits. Comprehensive implementation details are provided in Appendix C.2.

As presented in Table 3, LDDM-G demonstrates substantial performance gains over the MGDM baseline across all evaluated tasks and model scales. For instance, **with the 85M parameter model, LDDM-G achieves a 16% improvement on Game of 24 and an almost 8% gain on Countdown 4.** The loopholing mechanism drives these improvements by preserving contextual ambiguity, rather than prematurely committing to a single token per step.

**Table 3:** Success rates (%) on the Countdown (CD) and Game of 24 (G24) tasks.

| Architecture | Params | CD 4 | G 24 | CD 5 |
|---|---|---|---|---|
| MGDM[1] | 6M | 45 | 12 | 5.9 |
| | 85M | 86.5 | 47 | 35.7 |
| LDDM-G (Ours) | 6M | **56.3** | **28** | **10.3** |
| | 85M | **94.4** | **63** | **41.3** |

This allows it to maintain a richer representation of the solution space, enabling a more effective exploration of the multiple pathways required in complex reasoning tasks and ultimately enhancing its capacity for structured, multi-step reasoning.

## 6.3 ABLATION STUDY

In this section, we investigate the effects of our design choices by conducting a series of ablation studies. Specifically, we analyze the performance variations with respect to different self-conditioning rates in training, assess the impact of propagating the continuous latent representation, and examine the changes in excessive oscillation upon applying the loopholing mechanism.

---

[1]Reported MGDM baselines may differ slightly from those in Ye et al. (2024) due to differences in evaluation criteria. Specifically, we consider solutions invalid if they use numbers not provided in the input or derived from previous expressions, which were not filtered under the original codebase.

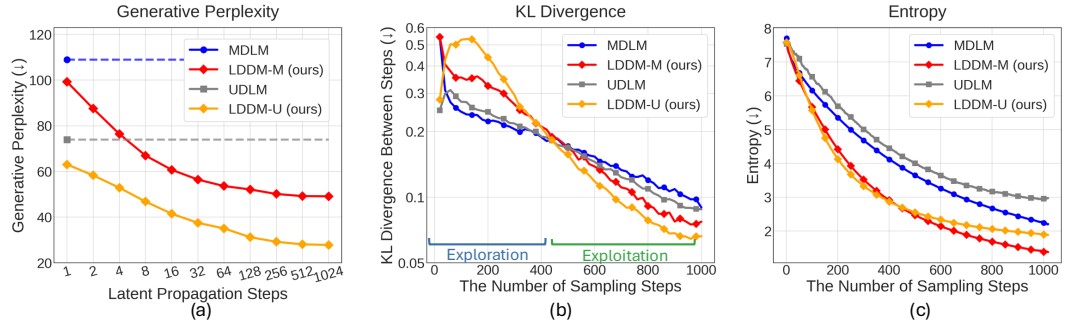

**Figure 5:** **(a)** Generative perplexity across varying latent propagation steps. **(b)** KL divergence (log-scale) between the predicted token distribution at each step $t$ and the distribution from 20 steps prior $(t-20)$ during the generation. **(c)** Entropy of the predicted token distributions throughout the generation.

**Self-Conditioning Rate**. We first evaluate the impact of varying the self-conditioning rate, denoted by $p$. We assess performance by measuring zero-shot perplexities on the datasets from Section 6.1, using models trained on the LM1B dataset. The results, presented in Fig. 4(c), show that **LDDM-M generally achieves its best performance across various unseen datasets when $p$ is set between $0.5$ and $0.9$.** This suggests that this range provides an effective balance, allowing the contextual latent representation to be robustly learned and utilized through the two-pass self-conditioning mechanism, with detailed results shown in Table 5.

**Latent Propagation Length**. We further investigate whether the efficacy of the loopholing mechanism accumulates over time. To see this, we assess the model's performance as a function of the latent propagation length $k$. Specifically, using a model trained on the OWT dataset, we generate samples with 1024 sampling steps and, every $k$ steps, reset the context latent to the self-conditioned latent instead of carrying it over from the previous step. This procedure limits the accumulation window, so larger $k$ means longer propagation. As shown in Fig. 5(a), performance improves as $k$ increases, suggesting that **the sustained propagation of accumulated latent information is effective in generating higher quality samples.**

**Idle Steps and Excessive Oscillation**. To investigate whether Loopholing mitigates excessive oscillation, we introduce two metrics—Temporal KL divergence (TKL) and Token-Prediction Entropy (TPE). For a sequence of length $L$ over $T$ number of sampling steps, these are defined as follows:

$$D_{\text{TKL}}(t) = \frac{1}{L}\sum_{\ell=1}^{L} D_{\text{KL}}\left(\mathbf{x}_{\theta,t+\frac{20}{T}}^{\ell} \| \mathbf{x}_{\theta,t}^{\ell}\right), \qquad H_{\text{TPE}}(t) = \frac{1}{L}\sum_{\ell=1}^{L} H(\mathbf{x}_{\theta,t}^{\ell}), \tag{9}$$

where $D_{\text{KL}}$ is the KL divergence and $H$ is the entropy. The TKL metric evaluates the rate of change in the token distribution across denoising steps (here measured with a 20-step lookback), whereas the TPE metric assesses the level of confidence or certainty in the model's predictions at each step. We therefore interpret a high TKL as reflecting faster progress along the denoising trajectory—closely tied to the *steps without progress* phenomenon—whereas a high TPE value serves as an indicator of excessive oscillatory behavior during generation.

We measure these metrics on models trained on OWT dataset with 1024 sampling steps. In Fig. 5(b), we first see an interesting crossover points in the middle where the behavior between LDDMs and non-LDDM-based models reverses. During the first half, the LDDM-based models show higher TKL than non-LDDM models. This means that **LDDMs update their predictions more actively at each denoising step.** We see this phase, where the model tries to search for the target topic to generate, as an exploration phase. Interestingly, the trend reverses during the second half of the denoising steps by showing lower TKL than non-LDDM models. This means, **LDDMs try to change more conservatively showing less oscillations.** As shown in Fig. 5(c), LDDMs maintain consistently lower token-level entropy. This indicates that **the stable contextual information carried by loopholing allows the model to make more confident and decisive predictions across the denoising trajectory.**

## 7 Discussion

**Computation and Memory.** While loopholing significantly improves the performance of discrete diffusion models, it also introduces certain limitations. Most notably, training requires about 30% more time compared to standard models, although it adds almost no overhead at inference time. In addition, doubling the embeddings to support both the sampling and contextual pathways leads to increased memory consumption. Moreover, the current training formulation considers only single-step updates, suggesting potential benefits from explicitly designing multi-step training strategies to better exploit long-range dependencies through the context latent path.

**Relations to Recurrent Neural Networks.** An insightful perspective on loopholing and LDDMs is to view them through the lens of Recurrent Neural Networks (RNNs) (Goodfellow et al., 2016). In this interpretation, the deterministic update path in loopholing corresponds to the hidden-state update of an RNN, while the stochastic and discrete outputs play the role of the RNN's output, which is then fed back as input at the next step—akin to an autoregressive RNN. However, the key difference lies in the training procedure: loopholing diffusion enables simulation-free training, whereas RNNs typically require rollout-based training. We believe that further exploring the connection between loopholing diffusion and RNNs would be a compelling direction for future work.

**General Limitations.** Furthermore, our current contribution is a novel architecture supported by empirical evidence. However, a rigorous mathematical framework that incorporates loopholing into the standard diffusion framework has not yet been developed, marking a natural direction for future theoretical work. In terms of scalability, our experiments have thus far been conducted on moderately sized models feasible within an academic setting, and extending loopholing to larger scales will be important for fully assessing its potential.

## 8 Conclusion

In this work, we identified the *sampling wall* as a key limitation of discrete diffusion models, where rich distributional information collapses into one-hot representations, leading to inefficiencies such as steps without progress and excessive oscillation. To overcome this, we proposed the loopholing mechanism and developed Loopholing Discrete Diffusion Models (LDDMs), which preserve and propagate distributional context latent across denoising steps through a deterministic latent pathway. Extensive experiments demonstrated that LDDMs improve fluency, naturalness, and semantic consistency in text generation and reasoning tasks, significantly narrowing the performance gap with autoregressive models. These results highlight loopholing as a general mechanism to enhance discrete diffusion, with promising future directions including multimodal extensions, theoretical understanding, and integration with broader non-autoregressive frameworks.

## Reproducibility statement

We provide detailed information on datasets, model configurations, training procedures, and evaluation protocols in Appendix C. Our source code and scripts for data processing, training, and evaluation are available at `https://github.com/ahn-ml/lddm`.

## Acknowledgment

This research was supported by Brain Pool Plus Program (No. 2021H1D3A2A03103645) and GRDC (Global Research Development Center) Cooperative Hub Program (RS-2024-00436165) through the National Research Foundation of Korea (NRF) funded by the Ministry of Science and ICT (MSIT). This research was also supported by the AI Computing Infrastructure Enhancement User Support Program (RQT-25-090190) funded by the Ministry of Science and ICT (MSIT), Republic of Korea. Justin Deschenaux is supported by the Swiss State Secretariat for Education, Research and Innovation (SERI). We acknowledge the SCITAS team at EPFL for providing access to their cluster, and the Swiss National Supercomputing Centre for the Alps platform. We thank the members of the Machine Learning and Mind Lab (MLML) and Lan Tran for valuable discussions and assistance.

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

## A  USE OF LARGE LANGUAGE MODELS

During the preparation of this manuscript, we utilized a large language model to improve grammar, clarity, and overall readability.

## B  UNIFORM DIFFUSION MODELS (UDMs)

In contrast to the masking approach of MDMs, Uniform Diffusion Models (UDMs) employ a uniform noising strategy (Austin et al., 2021; Schiff et al., 2024). In UDMs, the prior distribution $\pi$ is a uniform distribution over the vocabulary, denoted as $\mathbf{u} = \mathbf{1}/K$, where $\mathbf{1}$ is a $K$-dimensional vector of all ones. This forward process gradually replaces an original token with a random token drawn uniformly from the vocabulary.

Similar to MDMs, generation is performed by approximating the clean data $\mathbf{x}$ in the true reverse posterior $q(\mathbf{z}_s|\mathbf{z}_t, \mathbf{x})$ with a neural network $\mathbf{x}_\theta(\mathbf{z}_t, t)$. The specific formulation for this approximated posterior is given by:

$$q(\mathbf{z}_s|\mathbf{z}_t, \mathbf{x}_\theta(\mathbf{z}_t, t)) = \mathrm{Cat}\left(\mathbf{z}_s; \frac{K\alpha_t \mathbf{z}_t \odot \mathbf{x}_\theta + (\alpha_{t|s} - \alpha_t)\mathbf{z}_t + (\alpha_s - \alpha_t)\mathbf{x}_\theta + \frac{(\alpha_s - \alpha_t)(1-\alpha_s)}{K\alpha_s}\mathbf{1}}{K\alpha_t\langle\mathbf{x}_\theta, \mathbf{z}_t\rangle + 1 - \alpha_t}\right),$$
(10)

where $\odot$ denotes the Hadamard product, $\alpha_{t|s} = \frac{\alpha_t}{\alpha_s}$, and $\mathbf{x}_\theta$ is shorthand for $\mathbf{x}_\theta(\mathbf{z}_t, t)$. Unlike MDMs, which fix generated tokens, UDMs enable iterative refinement of the entire sequence.

For training, UDMs also optimize the continuous-time formulated NELBO. The training objective takes the following form (Schiff et al., 2024):

$$\mathcal{L}_{\mathrm{NELBO}} = \mathbb{E}_{t\sim\mathcal{U}[0,1],\mathbf{z}_t\sim q(\mathbf{z}_t|\mathbf{x})}\left[\frac{\alpha_t'}{K\alpha_t}\left[\frac{K}{\bar{\mathbf{x}}_i} - \frac{K}{(\bar{\mathbf{x}}_\theta)_i} - \sum_j \frac{\bar{\mathbf{x}}_j}{\bar{\mathbf{x}}_i}\log\frac{(\bar{\mathbf{x}}_\theta)_i \cdot \bar{\mathbf{x}}_j}{(\bar{\mathbf{x}}_\theta)_j \cdot \bar{\mathbf{x}}_i}\right]\right],$$
(11)

where $(\mathbf{x})_j$ is the $j$-th element of a vector $\mathbf{x}$, $\bar{\mathbf{x}} = K\alpha_t\mathbf{x} + (1-\alpha_t)\mathbf{1}$, $\bar{\mathbf{x}}_\theta = K\alpha_t\mathbf{x}_\theta + (1-\alpha_t)\mathbf{1}$, and $i = \mathrm{argmax}_{j\in[K]}(\mathbf{z}_t)_j$ is the index of the non-zero entry in $\mathbf{z}_t$. Recently, Duo (Sahoo et al., 2025) proposed a low-variance objective for UDMs with improved empirical performance.

## C  EXPERIMENT DETAILS

### C.1  LANGUAGE MODELING

#### C.1.1  EXPERIMENT DETAILS

**MDLM and UDLM settings.**  For our implementation, we followed best practices from prior work: we employed 64-bit precision for MDLM to ensure more accurate categorical sampling (Zheng et al., 2024), and we adopted the loss implementation from (Sahoo et al., 2025) for UDLM to improve its numerical stability.

**LM1B.**  For the One Billion Word Dataset (LM1B) (Chelba et al., 2013), we used the `bert-base-uncased` tokenizer (Devlin et al., 2019) with a fixed context length of 128 tokens. Sequences shorter than this were handled with padding. The model architecture is based on the Diffusion Transformer (DiT) (Peebles & Xie, 2023) with rotary embeddings (Su et al., 2024). The model consists of 12 Transformer blocks, each with 12 attention heads and a hidden dimension of 768. A dropout rate of 0.1 was applied throughout the model.

For optimization, we used the Adam optimizer (Kingma & Ba, 2014) with a learning rate of 3e-4, betas of (0.9, 0.999), and an epsilon of 1e-8. The learning rate was linearly warmed up from 0 to 3e-4 over the first 2,500 steps. Training was conducted for 1M steps on 8 NVIDIA RTX 4090 GPUs. The global batch size was set to 512, which was achieved by assigning a batch size of 32 to each GPU and applying gradient accumulation over 2 steps. Additionally, we applied an Exponential Moving Average (EMA) with a rate of 0.9999 and gradient clipping with a threshold of 1.0. For LDDMs, the self-conditioning rate was set to $p = 0.9$, and to stabilize early training we initialized layer normalization parameters to $\beta = \gamma = 0$.

**OWT.** For the OpenWebText (OWT) dataset (Gokaslan & Cohen, 2019), we used the `gpt2` tokenizer (Radford et al., 2019) with a context length of 1024 tokens. To maximize context utilization, we employ sentence packing during preprocessing (Austin et al., 2021). The model architecture and hyperparameters are largely similar to the LM1B experiment. Specifically, it is based on a DiT with rotary embeddings and includes 12 Transformer blocks, 12 attention heads, a hidden dimension of 768, and a dropout rate of 0.1.

Optimization was performed using the Adam optimizer (learning rate 3e-4, betas=(0.9, 0.999), epsilon=1e-8), with a linear learning rate warm-up over the initial 2,500 steps. The model was trained for 1M steps using 16 H100 GPUs. The global batch size was 512, configured by assigning a batch size of 32 to each GPU. Similar to the LM1B setup, we applied an EMA with a rate of 0.9999 and gradient clipping with a threshold of 1.0. LDDMs used a self-conditioning rate of 0.9, and we zero-initialized LayerNorm ($\beta = \gamma = 0$) to stabilize early training.

**Time Conditioning.** To remain faithful to prior work, we adopt the canonical setting used by each baseline: MDLM and LDDM-M are trained without time conditioning following Sahoo et al. (2024); SEDD (Lou et al., 2023), UDLM (Schiff et al., 2024), and LDDM-U use time conditioning as in their original implementations. We treat time conditioning as orthogonal to our contribution—loopholing adds a deterministic latent pathway and can be combined with either setting—so we do not tune it beyond faithfully reproducing baselines.

### C.1.2 PERPLEXITY DETAILS

In discrete diffusion models, we approximate perplexity (PPL) using the Negative Evidence Lower Bound (NELBO; Eqn. 3). The perplexity for a sequence of length L, $\mathbf{x}_{1:L}$, is defined and upper-bounded as follows:

$$\text{PPL}(\mathbf{x}_{1:L}) = \exp\left(-\tfrac{1}{L}\sum_{i=1}^{L}\log p(\mathbf{x}_i \mid \mathbf{x}_{<i})\right) = \exp\left(-\tfrac{1}{L}\log p(\mathbf{x}_{1:L})\right) \leq \exp\left(\tfrac{1}{L}\text{NELBO}(\mathbf{x}_{1:L})\right).$$

$$(12)$$

Here, the marginal log-likelihood $\log p(\mathbf{x})$ is intractable to compute directly. We leverage its relationship with the Negative Evidence Lower Bound (NELBO), where $-\log p(\mathbf{x}) \leq \text{NELBO}(\mathbf{x})$. This relationship allows us to use the computable upper bound as our perplexity metric. Since the NELBO is computed via a single Monte Carlo estimation over time steps $t$, which can be stochastic and exhibit high variance. To reduce the variance of this estimation, we adopt the low-discrepancy sampling technique from MDLM (Sahoo et al., 2024), which ensures that sampled time steps for each batch are more evenly spaced across the time interval [0,1].

Even with this improvement, the final value can be influenced by factors like batch size and hardware. Therefore, to ensure a fair and consistent evaluation, we compute all perplexity scores using a fixed experimental setup: two NVIDIA RTX 4090 GPUs with a batch size of 16 per GPU, under an identical software environment. (same PyTorch version, CUDA version, and library configurations).

### C.1.3 DATASETS FOR ZERO-SHOT LIKELIHOOD EVALUATION

We evaluate our model's zero-shot likelihood on a diverse suite of standard benchmarks. The datasets include the Penn Treebank (PTB) (Marcus et al., 1993), Wikitext (Merity et al., 2016), Lambada (Paperno et al., 2016), AG News (Zhang et al., 2015), and a corpus of scientific articles from Pubmed and Arxiv (Cohan et al., 2018).

### C.1.4 GENERATION QUALITY EVALUATION

**Generative Perplexity.** We evaluate the quality of generated text using generative perplexity (Gen PPL), computed with a pretrained GPT-2 Large model (Radford et al., 2019). Given a generated sequence of length $L$ composed of discrete tokens $\mathbf{x}^{(1:L)}$, the perplexity is calculated as:

$$\exp\left(-\frac{1}{L}\sum_{i=1}^{L}\log p_\phi(\mathbf{x}^{(i)} \mid \mathbf{x}^{(<i)})\right).$$

$$(13)$$

This metric reflects how likely the GPT-2 large model considers the sample, providing a proxy for overall sample quality. For this evaluation, we generate 512 samples using the model trained on OpenWebText (OWT) dataset and report the average perplexity across all samples.

**Sentence Entropy.** As shown in Zheng et al. (2024), Gen PPL can be deceptively low when generations have very low sentence entropy. To check for this, we measure sentence entropy for each sample. Sentence entropy indicates how many diverse tokens are used within a sample.

For a single generated sample of length $L$, let $\text{count}(v)$ be the number of times token $v$ appears. The sentence entropy for that sample is:

$$-\sum_{v \in \mathcal{V}} \frac{\text{count}(v)}{L} \log \left( \frac{\text{count}(v)}{L} \right), \tag{14}$$

where $\mathcal{V}$ is the vocabulary. We generate 512 samples and report the average sentence entropy.

**G-eval Scoring.** We evaluate the quality of generated texts using the LLM scoring framework (Liu et al., 2023), with GPT-4.1 serving as the evaluator. For each model trained on OpenWebText (Gokaslan & Cohen, 2019) dataset, we sample 512 unconditional generations. Due to sentence packing during training, each generation might contain multiple sequences separated by the end-of-sequence token ([EOS]). For evaluation, we retain only the first sequence.

Each sequence is independently rated by GPT-4.1 based on two criteria:

- **Consistency** (1–10): Evaluates whether the generated text maintains a coherent topic without contradictions or context shifts.
- **Naturalness** (1–10): Assesses grammar, fluency, idiomatic usage, and freedom from spelling or punctuation errors.

Scores are assigned in a zero-shot manner using the prompt provided in Fig. 6. To improve the accuracy of the measurements, we performed four evaluations for each sequence with the temperature set to 1.0 and assigned the average of these scores. We report the average score across all 512 sequences as the final measure of model quality.

## C.2 Reasoning Task

**Datasets.** We evaluate our method on two arithmetic reasoning tasks that require multi-step reasoning: Countdown (Gandhi et al., 2024) and Game of 24 (Yao et al., 2023). The Countdown task requires the model to use a given set of numbers and basic arithmetic operations (addition, subtraction, multiplication, and division) to reach a specific target number. For instance, in Countdown4, given the input numbers $\{24, 59, 23, 77\}$ and a target of 29, a valid solution is to first calculate $24 + 59 = 83$ and $77 - 23 = 54$, and then use these intermediate results to reach the target with $83 - 54 = 29$. Countdown5 extends this task to five input numbers, while the Game of 24 is a variant of Countdown4 where the target is always 24. We use the datasets released by Ye et al. (2024) without additional filtering or preprocessing.

**Setup.** Our experimental setup follows that of the Multi-Granularity Diffusion Model (MGDM) (Ye et al., 2024). We use the MGDM as our base architecture and integrate the loopholing mechanism to create what we call LDDM-G. MGDM extends the standard discrete diffusion objective with an adaptive token-level reweighting term and an easy-first TopK decoding strategy at inference.

Regarding TopK decoding, we made a notable adjustment to the original implementation. The original MGDM recalculates probabilities over all tokens (both masked and unmasked) at each step, which permits overwriting already generated tokens—a form of remasking. This approach conflicts with the training objective, which focuses exclusively on predicting masked tokens. Therefore, following prior work (Nie et al., 2025; Kim et al., 2025), we modify the decoding process to keep unmasked tokens fixed and apply the uncertainty-based TopK selection only to masked positions.

However, Table 4 shows that the original MGDM-style TopK decoding leads to slight performance gains. This is noteworthy because the method relies on distributions over unmasked tokens, for which it was not explicitly trained.

You will be given one piece of text.
Your task is to rate the text on two metrics. Please read and understand these instructions carefully, and keep this document open while reviewing.

Evaluation Criteria

Consistency (1–10)
A high score (10) indicates that the text maintains a single, coherent context throughout.
A low score (1) is given if the text shifts topic, contradicts itself, or loses logical flow.

Naturalness (1–10)
A high score (10) means the text is grammatically correct, idiomatic, and free of spelling or punctuation errors.
A low score (1) is given if the text contains frequent grammar mistakes, awkward phrasing, or typos.

Evaluation Steps

1. Read the generated text carefully and identify its intended context and message.
2. For Consistency, ask yourself:
Does the text stay on topic without introducing unrelated ideas?
Are there any contradictions or abrupt shifts in meaning?
3. For Naturalness, ask yourself:
Is the writing grammatically sound and easy to read?
Are phrases idiomatic, and is punctuation used correctly?
4. Assign each metric a score from 1 (lowest) to 10 (highest) based on the above definitions.

Text:
`{sample}`

Evaluation Form (scores ONLY):
– Consistency:
– Naturalness:

**Figure 6:** G-Eval prompt template used with GPT-4.1. `{sample}` is replaced with the generated sequence to evaluate.

**Table 4: Performance using original MGDM TopK decoding.** Success rates (%) on Countdown and Game of 24 tasks using the original MGDM TopK decoding implementation, which includes probabilities over unmasked tokens. Numbers in parentheses indicate accuracy improvements from using MGDM TopK decoding

| Architecture | Params | CountDown4 | Game of 24 | CountDown5 |
|---|---|---|---|---|
| MGDM (Retrained) | 6M | 47.6 (+2.6) | 12 | 7.6 (+1.7) |
| | 85M | 87.1 (+ 0.6) | 52 (+5) | 36.9 (+1.2) |
| LDDM-G (Ours) | 6M | **57** (+0.7) | **29** (+1) | **11.6** (+1.3) |
| | 85M | **94.5** (+0.1) | **64** (+1) | **42.6** (+1.3) |

**Training.** Both MGDM and LDDM-G models are trained for 600 epochs with a batch size of 1024. We use the Adam optimizer (Kingma & Ba, 2014) with betas of (0.9, 0.999) and an epsilon of 1e-8. For the 6M models, we use a learning rate of 1e-3, and for the 85M models, we use a learning rate of 3e-4; a cosine learning rate schedule is employed for both. All models are trained on 8 RTX 4090 GPUs. The model architecture is based on GPT-2 (Radford et al., 2019), without the causal mask for bidirectional attention. MGDM is trained with hyperparameters $\alpha = 0.25$ and $\beta = 2$. For LDDM-G, we apply self-conditioning with $p = 0.9$.

**Evaluation.** A generation is considered correct if the resulting arithmetic expression evaluates exactly to the target number without reusing inputs or generating invalid intermediate values. Our evaluation script employs stricter criteria than the original MGDM implementation. Specifically, the original evaluation only verified that each intermediate equation and the final result were mathematically valid, without penalizing generations using numbers not provided in the input or not derived

> There was no way he would come here on his own.
> He ordered a cup of coffee, and then we just sat in silence.
> "So," Aidan finally said, "How's it going?"
> I laughed. "Not much has changed since the last time I saw you."
> "Ya know, you eat here a lot," said **Aidan**

**Figure 7:** An example from the LAMBADA dataset. The goal is to predict the final word, "Aidan".

from previous expressions. We enhance this by explicitly filtering out generations using such invalid numbers, ensuring faithful and input-grounded reasoning.

# D  ADDITIONAL RESULTS

## D.1  DETAILED RESULTS FOR SELF-CONDITIONING RATE

**Table 5:** Perplexities ($\downarrow$) of SEDD Absorb, MDLM, and our LDDM-M with varying self-conditioning rates $p$. The model is trained with diverse $p$ values and evaluated with $p = 1.0$. All scores are reported as upper-bound estimates.

|  | LM1B(trained) | PTB | Wikitext | Lambada | AG News | Pubmed | Arxiv |
|---|---|---|---|---|---|---|---|
| SEDD Absorb | 28.39 | 108.63 | 78.61 | 99.44 | 61.57 | 75.09 | 142.19 |
| MDLM | 27.60 | 110.90 | 74.43 | 100.11 | 60.50 | 70.72 | 140.62 |
| LDDM-M (ours) | | | | | | | |
| - $p = 1.0$ | 26.34 | 101.92 | 70.51 | 92.17 | 57.71 | 69.25 | 140.29 |
| - $p = 0.9$ | **25.95** | **99.92** | 66.87 | **89.62** | 56.72 | 67.38 | 136.04 |
| - $p = 0.7$ | 26.14 | 102.75 | **66.55** | 90.64 | **56.43** | 67.09 | 134.73 |
| - $p = 0.5$ | 26.39 | 100.12 | 66.58 | 89.63 | 57.99 | **67.01** | **128.06** |
| - $p = 0.3$ | 26.66 | 101.88 | 71.90 | 90.16 | 58.46 | 68.08 | 135.33 |
| - $p = 0.1$ | 26.88 | 100.52 | 69.02 | 90.38 | 59.10 | 69.69 | 136.25 |

## D.2  DOWNSTREAM TASKS

We evaluated MDLM and LDDM-M on various downstream tasks using the `lm-eval-harness` library (Gao et al., 2024). Both models were pre-trained on OpenWebText. Our evaluation includes six multiple-choice tasks—ARC-Easy and ARC-Challenge (Clark et al., 2018), HellaSwag (Zellers et al., 2019), MathQA (Amini et al., 2019), PIQA (Bisk et al., 2020), and WinoGrande (Sakaguchi et al., 2019)—and one generation task, LAMBADA (Paperno et al., 2016). Since the library is designed for autoregressive models, we adapted the approach for the masked diffusion framework.

**Multiple-choice tasks.** Following Deschenaux et al. (2025), we adapted the evaluation for MDLMs. Autoregressive models select the answer with highest log-likelihood via $\arg\max_i \log p(\mathbf{y}_i|\mathbf{x})$ where $\mathbf{x}$ is the context and $\mathbf{y}_i$ is an answer option. However, MDLMs model the joint probability $\log p(\mathbf{x}, \mathbf{y}_i)$ of the entire sequence, requiring a different approach.

Bayes' rule provides a solution by connecting conditional and joint probabilities:

$$\log p(\mathbf{y}_i|\mathbf{x}) = \log p(\mathbf{x}, \mathbf{y}_i) - \log p(\mathbf{x}) \propto \log p(\mathbf{x}, \mathbf{y}_i) \tag{15}$$

Since $\log p(\mathbf{x})$ is constant across all answer options, ranking by joint probability is equivalent to ranking by conditional probability. We bound the joint probability using NELBO, compute Monte Carlo estimates, and select the answer with the lowest NELBO.

**LAMBADA task.** Unlike multiple-choice tasks where models select from given options, LAMBADA requires generating the last word and comparing it with the ground truth. During evaluation, we identified a critical issue with the dataset format. As shown in Fig. 7, the final word lacks any terminal punctuation (period, question mark, etc.). While this poses no problem for autoregressive models that only consider preceding context, it creates a significant challenge for discrete diffusion

**Table 6:** Performance on downstream tasks. LDDM-M substantially outperforms MDLM on the generation task (LAMBADA) while showing comparable accuracy on multiple-choice benchmarks.

| Model | LAMBADA | ARC-e | ARC-c | HSwag | MathQA | PIQA | WinoG |
|-------|---------|-------|-------|-------|--------|------|-------|
| MDLM | 40.46 | **36.49** | **25.17** | 31.81 | 21.51 | 57.62 | **51.85** |
| LDDM-M | **52.40** | 36.03 | 23.21 | **33.11** | **22.21** | **58.16** | 51.46 |

**Table 7:** Impact of adding an MLP layer on perplexity across benchmarks. All reported values are upper bounds.

| | LM1B(trained) | PTB | Wikitext | Lambada | AG News | Pubmed | Arxiv |
|---|---------------|-----|----------|---------|---------|--------|-------|
| MDLM | 27.60 | 110.90 | 74.43 | 100.11 | 60.50 | 70.72 | 140.62 |
| + MLP | 27.21 | 107.96 | 74.61 | 97.04 | 59.10 | 70.31 | 139.42 |
| LDDM-M (ours) | 25.95 | 99.92 | **66.87** | 89.62 | 56.72 | 67.38 | 136.04 |
| + MLP | **25.87** | **99.54** | 66.96 | **89.33** | **56.43** | **66.05** | **127.71** |

models. When the input is formatted as `... said [MASK] [EOS]`, the EOS token signals the end of sequence, causing the model to generate punctuation marks to properly terminate the sentence rather than predicting the target word "Aidan".

To resolve this issue, we modified the model input by inserting an additional `[MASK]` token before the `[EOS]` token. This extra position serves as a placeholder for terminal punctuation, allowing the model to correctly predict the target word in the original mask position. The added token is used only during inference and is not included in evaluation. For targets spanning multiple tokens, we adopted the iterative decoding implementation from Nie et al. (2024), where we unmask one token at a time based on confidence. These modifications resulted in improved performance and more reliable evaluation.

**Results.** Table 6 shows that LDDM-M achieves notably higher accuracy on the generation task (LAMBADA) with 52.40 vs 40.46, while maintaining comparable performance on the likelihood-based multiple-choice tasks.

### D.3 DETERMINISTIC PATH AUGMENTATION

To understand the effect of incorporating additional parameters into the deterministic path, we investigate augmenting the deterministic path with a two-layer MLP featuring an expansion ratio of 4. In this modification, we update the latent embeddings as $\mathbf{h}'_t = \mathbf{h}_t + \mathrm{MLP}(\mathbf{h}_t)$, followed by layer normalization to produce the latent representation $\mathbf{e}_t = E_\theta(\mathbf{z}_t) + \mathrm{LN}(\mathbf{h}'_t)$. The original MDLM architecture applies a similar MLP directly to the token embeddings, where $\mathbf{z}'_t = E_\theta(\mathbf{z}_t) + \mathrm{MLP}(E_\theta(\mathbf{z}_t))$ and the latent representation is computed as $\mathbf{e}_t = E_\theta(\mathbf{z}_t) + \mathrm{LN}(\mathbf{z}'_t)$. The subsequent structure is the same as before (see Section 3.1). The model was trained on the LM1B dataset.

As shown in Table 7, this augmentation yields only marginal performance gains. This finding suggests that the primary benefit of our framework stems from contextual latent propagation across time steps, rather than from additional parametric complexity in the deterministic path.

### D.4 APPLICATION OF LOOPHOLING TO AUTOREGRESSIVE MODELS

To investigate the broader applicability of our proposed mechanism, we adapted loopholing for a standard autoregressive language model. During training, the model performs two forward passes. The first pass generates a pseudo-context from the input sequence. This pseudo-context is then shifted by one position to the right (with the first position embedding being a zero vector), and fed

**Table 8:** Unconditional Gen PPL on 512 samples (1024 sampling steps).

| Model | Gen PPL (↓) |
|-------|-------------|
| AR | 34.40 |
| AR-Loopholing | **34.21** |

**Table 9:** Perplexities (↓) of Autoregressive models trained on OpenWebText.

|  | OWT (trained) | PTB | Wikitext | LM1B | Lambada | AG News | Pubmed | Arxiv |
|---|---|---|---|---|---|---|---|---|
| AR (Baseline) | 17.27 | 118.39 | 34.88 | 55.91 | 48.22 | 60.60 | 42.72 | 46.40 |
| AR-Loopholing | **16.57** | **115.99** | **32.98** | **55.19** | **45.00** | **57.01** | **41.23** | **45.65** |

**Table 10:** Perplexities (↓) of models with matched computational budget, trained on OpenWebText. All reported perplexity values are upper bounds.

|  | OWT(trained) | PTB | Wikitext | LM1B | Lambada | AG News | Pubmed | Arxiv |
|---|---|---|---|---|---|---|---|---|
| MDLM (1M steps) | 23.05 | 86.33 | 36.30 | 66.73 | 48.36 | 68.62 | 41.94 | 37.52 |
| MDLM (2M steps) | 22.54 | **84.40** | 35.27 | **65.68** | 48.07 | 66.26 | 42.08 | 36.75 |
| LDDM-M (1M steps) | **21.90** | 85.80 | **33.27** | 69.53 | **44.22** | **62.55** | **39.74** | **34.96** |

into the second forward pass along with the original token embeddings. This setup allows each token's prediction to be conditioned on the continuous representation of the preceding token from the initial pass. During inference, the model generates tokens sequentially, passing both the generated token and its corresponding latent embedding to the next step, a technique similar to another existing method (Zhuang et al., 2025).

However, this approach did not meaningfully improve sample quality (Table 8), despite a slight improvement in perplexity scores (Table 9). We attribute this outcome to the fundamental differences in their generation processes. In discrete diffusion, the sampling wall problem is more severe because the loss of the predicted distribution occurs for many tokens at every step of the iterative refinement. In contrast, autoregressive models operate with a fixed context, stably predicting only one token at a time. Consequently, the information loss from a single sampling step is less significant, and the primary challenges loopholing is designed to mitigate are less prevalent in this framework.

### D.5    IMPACT OF MULTISTEP BACKPROPAGATION

In our standard self-conditioning setup, gradients are backpropagated only through the second forward pass for computational efficiency. To see if allowing gradients to propagate through more steps would be beneficial, we conducted an experiment. Since training from scratch is computationally prohibitive, we performed a short fine-tuning experiment for 10K steps on a model pre-trained on the OpenWebText dataset for 1M steps with self-conditioning. In this fine-tuning stage, we extended the training process to three and four forward passes, including the sampling steps between them to mimic the actual generation process with 1024 denoising steps. Gradients from the final pass were allowed to flow back to the second pass, while the initial pseudo-context generation pass remained detached.

**Table 11:** Gen PPL on 512 samples (1024 sampling steps) after 10K fine-tuning steps.

| Model | Gen PPL (↓) |
|---|---|
| LDDM-M | 49.13 |
| + 3 Fwd Pass (10K) | 83.08 |
| + 4 Fwd Pass (10K) | 60.07 |

As shown in Table 11, this approach led to a decrease in performance, as measured by gen PPL. We hypothesize this is because in the standard two-pass setup, the second pass learns to robustly refine any context from the detached first pass, since it cannot control its input. However, in the third and fourth passes, the model can influence the context it receives from the previous step. This likely leads to the model learning dependencies on specific context patterns, which harms its ability to generalize during inference and degrades generation quality.

### D.6    COMPARISON WITH MATCHED COMPUTATIONAL BUDGET

The loopholing mechanism requires two forward passes during each training step. While total computational cost (FLOPs) also includes backpropagation, we naively matched the number of forward passes as a proxy for the overall budget. To ensure LDDMs' performance gains stem from the deterministic path and not simply the increased computation, we ran a controlled experiment.

**Table 12:** Comparison of Sample Quality between Masked Diffusion and Autoregressive Models. All metrics were evaluated on 512 samples generated from models trained on OpenWebText. Metrics from the OpenWeb-Text validation set are also provided for reference

| | OWT Dataset Validation (512 samples) – Gen PPL: 14.88    Entropy: 5.442    Self-BLEU: 0.2498 | | | | | | | | |
| | Autoregressive Model (T=1024) – Gen PPL: 34.40    Entropy: 5.573    Self-BLEU: 0.2450 | | | | | | | | |

| | SEDD | | | MDLM | | | LDDM-M | | |
|---|---|---|---|---|---|---|---|---|---|
| T | Gen PPL | Entropy | Self-BLEU | Gen PPL | Entropy | Self-BLEU | Gen PPL | Entropy | Self-BLEU |
| 32 | 195.96 | 5.731 | 0.1971 | 198.72 | 5.745 | 0.1896 | 248.02 | 5.739 | 0.1673 |
| 64 | 143.04 | 5.681 | 0.2100 | 145.48 | 5.704 | 0.2051 | 122.34 | 5.674 | 0.2014 |
| 128 | 123.52 | 5.658 | 0.2153 | 125.04 | 5.681 | 0.2078 | 83.10 | 5.641 | 0.2156 |
| 256 | 114.12 | 5.643 | 0.2178 | 114.05 | 5.659 | 0.2126 | 64.76 | 5.600 | 0.2241 |
| 512 | 111.18 | 5.629 | 0.2178 | 112.83 | 5.652 | 0.2089 | 53.56 | 5.569 | 0.2289 |
| 1024 | 109.05 | 5.629 | 0.2214 | 108.94 | 5.637 | 0.2132 | 49.13 | 5.545 | 0.2301 |
| 2048 | 108.16 | 5.625 | 0.2168 | 107.20 | 5.625 | 0.2096 | 44.51 | 5.518 | 0.2295 |

**Table 13:** Sample Quality of Uniform Diffusion Based Models. All metrics were evaluated on 512 samples generated from models trained on OpenWebText.

| | UDLM | | | LDDM-U | | |
|---|---|---|---|---|---|---|
| T | Gen PPL | Entropy | Self-BLEU | Gen PPL | Entropy | Self-BLEU |
| 32 | 95.90 | 5.591 | 0.2447 | 75.55 | 5.551 | 0.2541 |
| 64 | 86.24 | 5.578 | 0.2436 | 56.27 | 5.538 | 0.2572 |
| 128 | 80.39 | 5.571 | 0.2407 | 45.47 | 5.518 | 0.2592 |
| 256 | 77.64 | 5.560 | 0.2423 | 38.76 | 5.494 | 0.2519 |
| 512 | 77.78 | 5.562 | 0.2367 | 32.83 | 5.447 | 0.2451 |
| 1024 | 73.95 | 5.547 | 0.2429 | 28.76 | 5.418 | 0.2343 |
| 2048 | 75.45 | 5.537 | 0.2380 | 25.06 | 5.366 | 0.2320 |

We trained a baseline MDLM for 2 million steps, thereby matching the number of forward passes of our LDDM-M trained for 1 million steps. The results confirmed that the LDDM-M (1M steps) significantly outperformed the MDLM (2M steps), as shown in Table 14 and Table 10. This demonstrates that the improvements achieved by loopholing are attributable to the propagation of contextual information, not a larger computational budget.

**Table 14:** Unconditional gen PPL on 512 samples (1024 sampling steps).

| Model | Gen PPL ($\downarrow$) |
|---|---|
| MDLM (1M steps) | 108.94 |
| MDLM (2M steps) | 108.22 |
| LDDM-M (1M steps) | **49.13** |

## D.7 DETAILED QUANTITATIVE ANALYSIS OF SAMPLE QUALITY

This section provides the specific numerical values for the sample quality analysis from Section 6.1, detailed in Table 12 and Table 13. We evaluate Generative Perplexity (Gen PPL), Sentence Entropy, and Self-BLEU. Self-BLEU (Zhu et al., 2018) is a metric that quantifies the internal diversity of a generated text corpus. For our experiments, we compute Self-BLEU scores using up to 4-grams with uniform weights (i.e., a weight of 0.25 for each n-gram from 1 to 4). Lower Self-BLEU scores indicate higher diversity, as they reflect less n-gram overlap among the generated samples. The autoregressive model used for comparison is a standard Transformer architecture. The analysis highlights that while LDDMs maintain contextual information about the target $\mathbf{x}$ across denoising steps, they also maintain generation diversity. We attribute this to the high diversity in the initial stages of generation, as suggested by our ablation study (Section 6.3).

## D.8 VISUALIZATION AND QUANTIFICATION OF IDLE STEPS

To verify the presence of idle steps in MDLM, we examine the number of tokens opened during a single generation trajectory. First, to visualize this phenomenon explicitly, we tracked the number of opened (updated) tokens at each denoising step for a single sample generated by MDLM with 1,024 sampling steps. As illustrated in Figure 8, a significant number of steps exhibit zero token changes. These instances correspond to idle steps, where the model fails to update any part of the sequence

despite expending computational resources. The scatter plot clearly shows clusters of zero-value points, confirming that idle steps are distinct events in individual trajectories.

To quantify the prevalence of this issue, we measured the average frequency of idle steps across 512 generated samples. We evaluated the model with varying total sampling steps ($T$). For $T = 256$, $512$, and $1024$, we observed an average of 4.8, 69, and 376 idle steps, respectively. These findings align with the observations reported in Chao et al. (2025), confirming that standard discrete diffusion models suffer from substantial computational inefficiency due to these non-progressive steps.

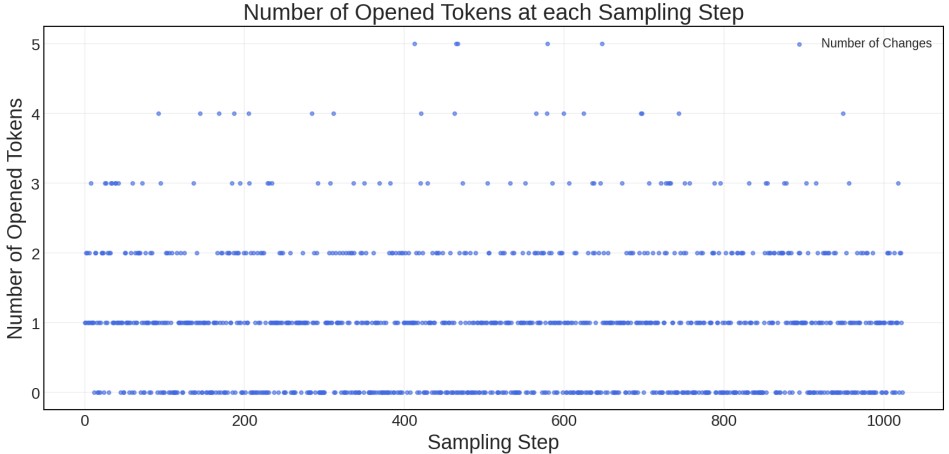

**Figure 8:** Number of opened tokens at each sampling step during a single generation process ($T = 1024$). The data points at $y = 0$ represent idle steps where no tokens were updated, highlighting the redundancy in the generation process.

### D.9 SCALABILITY ANALYSIS

To examine whether the improvements from Loopholing hold beyond our main configuration, we trained models at a moderately larger scale (169M → 424M; Depth: 24, Heads: 16, Hidden Dim: 1024). Both MDLM and LDDM-M were trained for 100K steps on OWT under identical settings.

**Table 15:** Perplexities (↓) at two model scales, trained for 100K steps on OpenWebText. All reported perplexity values are upper bounds.

| Model | Size | OWT | LM1B | LAMBADA | PTB | AG News | Pubmed | Arxiv | Wikitext |
|---|---|---|---|---|---|---|---|---|---|
| MDLM | 169M | 27.20 | 80.10 | 55.45 | 116.54 | 89.00 | 51.47 | 44.27 | 45.46 |
| MDLM | 424M | 22.49 | 68.55 | 47.49 | 78.67 | 64.68 | 40.98 | 36.15 | 34.90 |
| LDDM-M (ours) | 169M | 26.14 | 80.00 | 49.94 | 102.67 | 79.84 | 47.58 | 40.59 | 39.93 |
| LDDM-M (ours) | 424M | 20.20 | 64.97 | 42.52 | 72.13 | 53.82 | 36.88 | 32.89 | 31.10 |

**Table 16:** Generative perplexity (↓) at 1024 sampling steps for two model scales.

| Model | Size | Gen PPL (↓) |
|---|---|---|
| MDLM | 169M | 108.94 |
| MDLM | 424M | 85.62 |
| LDDM-M (ours) | 169M | 49.13 |
| LDDM-M (ours) | 424M | **40.10** |

As shown in Tables 15 and 16, LDDM-M consistently outperforms MDLM at both scales. At 424M parameters, LDDM-M achieves a Gen PPL of 40.10 compared to MDLM's 85.62, maintaining a roughly $2.1\times$ improvement. The consistent gap across the two configurations indicates that the loopholing mechanism does not diminish with increased model capacity.

## D.10 CONTINUED PRE-TRAINING WITH LOOPHOLING

To assess whether Loopholing can be applied to existing pretrained diffusion models without training from scratch, we start from a standard MDLM checkpoint trained for 1M steps on OWT and integrate the Loopholing mechanism by attaching zero-initialized layer normalization layers. We then continue pre-training with the LDDM-M self-conditioning objective for up to 200K steps.

**Table 17:** Perplexities ($\downarrow$) of continued pre-training from an MDLM (1M step) checkpoint. All reported perplexity values are upper bounds.

| Model | OWT | LM1B | LAMBADA | PTB | AG News | Pubmed | Arxiv | Wikitext |
|---|---|---|---|---|---|---|---|---|
| MDLM (1M step) | 23.05 | 66.73 | 48.36 | 86.33 | 68.62 | 41.94 | 37.52 | 36.30 |
| LDDM-M (1M step) | 21.90 | 69.53 | **44.22** | 85.80 | 62.55 | 39.74 | **34.96** | 33.27 |
| MDLM → LDDM-M (10K) | 22.39 | 65.83 | 45.70 | 83.45 | 61.70 | 40.57 | 36.55 | 33.92 |
| MDLM → LDDM-M (50K) | 21.91 | 65.34 | 45.09 | 82.20 | 60.53 | 39.22 | 35.59 | 33.35 |
| MDLM → LDDM-M (100K) | 21.71 | 64.57 | 44.44 | 80.77 | 59.83 | **38.97** | 35.35 | 33.02 |
| MDLM → LDDM-M (200K) | **21.50** | **64.42** | 44.27 | **80.04** | **59.44** | 39.10 | 35.14 | **32.83** |

**Table 18:** Generative perplexity ($\downarrow$) of continued pre-training across different sampling steps.

| Model | $T$=256 | $T$=512 | $T$=1024 | $T$=2048 |
|---|---|---|---|---|
| MDLM (1M step) | 114.05 | 112.83 | 108.94 | 107.20 |
| LDDM-M (1M step) | **64.76** | **53.56** | **49.13** | **44.51** |
| MDLM → LDDM-M (10K) | 91.04 | 82.45 | 78.31 | 74.65 |
| MDLM → LDDM-M (50K) | 75.73 | 66.57 | 61.96 | 56.69 |
| MDLM → LDDM-M (100K) | 71.01 | 60.72 | 57.20 | 51.59 |
| MDLM → LDDM-M (200K) | 66.06 | 55.81 | 51.51 | 47.04 |

The adapted model surpasses the original MDLM (1M) within only 10K continued pre-training steps in most benchmarks. After 200K steps, it closely approaches the performance of LDDM-M trained from scratch for 1M steps. In terms of Gen PPL at 1024 sampling steps, continued pre-training reduces it from 108.94 to 51.51, nearly matching the scratch-trained LDDM-M (49.13). These results confirm that Loopholing can be effectively integrated into existing diffusion model checkpoints through continued pre-training.

## D.11 ISOLATING SELF-CONDITIONING FROM LATENT PROPAGATION

To isolate the contribution of our recurrent latent path from the effect of self-conditioning, we compare three variants: (a) standard MDLM, (b) LDDM-M with latent propagation (our method), and (c) a variant that propagates the sampled token (drawn from $\mathbf{x}_\theta$) to the next step instead of the pre-sampling latent state $\mathbf{h}_t$, which is closer to standard self-conditioning (Chen et al., 2022). All models were trained for 200K steps on OpenWebText with the same configuration.

**Table 19:** Generative perplexity ($\downarrow$) at 1024 sampling steps, comparing latent propagation with sampled token propagation. All models trained for 200K steps on OpenWebText.

| Model (169M, 200K steps) | Gen PPL ($\downarrow$) |
|---|---|
| MDLM | 103.52 |
| LDDM-M (sampled token) | 93.71 |
| LDDM-M (latent) | **55.04** |

While propagating the sampled token offers a slight improvement over MDLM, it lags significantly behind latent propagation. This confirms that preserving pre-sampling information via a continuous latent path is the primary driver of Loopholing's improvements, rather than the self-conditioning training procedure itself.

## D.12 RESULTS UNDER ORIGINAL MGDM EVALUATION SCHEME

For transparency, we additionally evaluate LDDM-G and MGDM under the original evaluation and decoding scheme of Ye et al. (2024).

**Table 20:** Success rates (%) under the original MGDM evaluation scheme of Ye et al. (2024).

| Model | Params | Countdown 4 | Game of 24 | Countdown 5 |
|---|---|---|---|---|
| MGDM (retrained) | 6M | 51.5 | 32 | 22 |
| | 85M | 88.1 | 62 | 52 |
| LDDM-G (ours) | 6M | 60.6 | 45 | 23.6 |
| | 85M | 95.3 | 82 | 55.2 |

The relative performance trend remains unchanged under the original evaluation scheme, and LDDM-G continues to outperform MGDM across all tasks and model scales.

## E IMPLEMENTATION PSEUDO-CODE

Here is the pseudo-code for the generation process and training step of Loopholing Discrete Diffusion Models (LDDMs) with self-conditioning. While the forward function is shared across all models, the generation and training procedures are presented for LDDM-M for clarity. For LDDM-U, the same procedure applies by replacing the posterior and training objective with Eqn. 10 and Eqn. 11, respectively.

---

**Algorithm 1** LDDMs Forward Function

---

1: **Require:** Current token sequence $\mathbf{z}_t^{(1:L)}$, previous latent context $\mathbf{h}_t^{(1:L)}$, diffusion timestep $t$.
2: **Parameters:** Token embedding layer $E_\theta$, backbone $f_\theta$, output projection $g_\theta$.
3: **Output:** Logits $\mathbf{o}_s^{(1:L)}$, updated latent context $\mathbf{h}_s^{(1:L)}$.

4: **function** LDDMFORWARD$_\theta(\mathbf{z}_t^{(1:L)}, \mathbf{h}_t^{(1:L)}, t)$
5:      $\mathbf{v}_t^{(1:L)} \leftarrow E_\theta(\mathbf{z}_t^{(1:L)})$                ▷ Embed token sequence.
6:      $\mathbf{e}_t^{(1:L)} \leftarrow \mathbf{v}_t^{(1:L)} + \text{LayerNorm}(\mathbf{h}_t^{(1:L)})$     ▷ Fuse input with normalized latent.
7:      $\mathbf{h}_s^{(1:L)} \leftarrow f_\theta(\mathbf{e}_t^{(1:L)}, t)$          ▷ Update memory state via backbone.
8:      $\mathbf{o}_s^{(1:L)} \leftarrow g_\theta(\mathbf{h}_s^{(1:L)})$            ▷ Project to vocabulary space.
9:      **return** $\mathbf{o}_s^{(1:L)}, \mathbf{h}_s^{(1:L)}$
10: **end function**

---

---

**Algorithm 2** LDDM-M Generation Process

---

1: **Require:** Total diffusion steps $T$, sequence length $L$, forward function LDDMFORWARD$_\theta$.
2: **Output:** A generated sequence.

3: $\mathbf{z}_1^{(1:L)} \leftarrow ([\text{MASK}], \dots, [\text{MASK}])$       $\triangleright$ Initialize a sequence of length $L$ with MASK tokens.
4: $\mathbf{h}_1^{(1:L)} \leftarrow \mathbf{0}$       $\triangleright$ Initialize the latent embedding to a zero vector.

5: **for** $i = T \rightarrow 1$ **do**
6:     $t \leftarrow i/T$
7:     $s \leftarrow (i-1)/T$
8:     $\mathbf{o}_s^{(1:L)}, \mathbf{h}_s^{(1:L)} \leftarrow \text{LDDMFORWARD}_\theta(\mathbf{z}_t^{(1:L)}, \mathbf{h}_t^{(1:L)}, t)$    $\triangleright$ Predict logits and update latent.
9:     $\mathbf{x}_\theta^{(1:L)}(\mathbf{z}_t^{(1:L)}, \mathbf{h}_t^{(1:L)}, t) = \text{Softmax}(\mathbf{o}_s^{(1:L)})$   $\triangleright$ Predicted distribution of the clean sequence.
10:     **for** $\ell = 1 \rightarrow L$ **do**       $\triangleright$ Element-wise sampling (independent across $\ell$).
11:        $\mathbf{z}_s^{(\ell)} \sim q\big(\mathbf{z}_s^{(\ell)} \mid \mathbf{z}_t^{(1:L)}, \mathbf{x}_\theta^{(\ell)}(\mathbf{z}_t^{(1:L)}, \mathbf{h}_t^{(1:L)}, t)\big)$     $\triangleright$ The posterior $q$ is defined in Eqn. 2.
12:     **end for**
13:     $\mathbf{z}_t^{(1:L)} \leftarrow \mathbf{z}_s^{(1:L)}$
14: **end for**

15: **return** $\mathbf{z}_t^{(1:L)}$

---

**Algorithm 3** LDDM-M Training Step with Self-Conditioning

---

1: **Require:** Clean data of length $L$, $\mathbf{x}^{(1:L)}$, noise schedule $\alpha_t$, self-conditioning rate $p \in [0,1]$, forward function LDDMFORWARD$_\theta$.
2: **Ensure:** Training loss $\mathcal{L}$.

3: $t \sim \mathcal{U}[0,1]$       $\triangleright$ Sample a random time step.
4: **for** $\ell = 1 \rightarrow L$ **do**       $\triangleright$ Each token is processed independently.
5:     $\mathbf{z}_t^{(\ell)} \sim q(\mathbf{z}_t^{(\ell)} | \mathbf{x}^{(\ell)})$       $\triangleright$ Sample a noised input via the forward process $q$ (Eqn. 1).
6: **end for**
7: $\mathbf{h}_{\text{cond}}^{(1:L)} \leftarrow \mathbf{0}$       $\triangleright$ Initialize the contextual latent to zero.

8: With probability $p$:
9:     $\_, \mathbf{h}_{\text{pseudo}}^{(1:L)} \leftarrow \text{LDDMFORWARD}_\theta(\mathbf{z}_t^{(1:L)}, \mathbf{0}, t)$
10:     $\mathbf{h}_{\text{cond}}^{(1:L)} \leftarrow \text{STOPGRAD}(\mathbf{h}_{\text{pseudo}}^{(1:L)})$       $\triangleright$ Update with the detached pseudo-context.

11: $\mathbf{o}^{(1:L)}, \_ \leftarrow \text{LDDMFORWARD}_\theta(\mathbf{z}_t^{(1:L)}, \mathbf{h}_{\text{cond}}^{(1:L)}, t)$
12: $\mathbf{x}_\theta^{(1:L)}(\mathbf{z}_t^{(1:L)}, \mathbf{h}_{\text{cond}}^{(1:L)}, t) \leftarrow \text{Softmax}(\mathbf{o}_s^{(1:L)})$
13: $\mathcal{L} \leftarrow \sum_\ell \mathbb{I}[\mathbf{z}_t^{(\ell)} = \mathbf{m}] \frac{\alpha_t'}{1-\alpha_t} \log \langle \mathbf{x}_\theta^{(\ell)}(\mathbf{z}_t^{(1:L)}, \mathbf{h}_{\text{cond}}^{(1:L)}, t), \mathbf{x}^{(\ell)} \rangle$       $\triangleright$ Calculate loss based on Eqn. 3

14: **return** $\mathcal{L}$

---

## F  SAMPLES

Figures 9–12 present unconditioned samples generated with 1,024 sampling steps from models trained on OpenWebText. We highlight topic shifts and incoherent sequences in the baseline samples. MDLM exhibits both failure modes, while UDLM maintains local grammar but still drifts across unrelated topics. In contrast, LDDM samples preserve topical consistency over longer spans.

⟨|**endoftext**|⟩ NA laner damage/mon slot degenerate, they should all tend to take longer than the average to achieve three splits in NA. Any midlaner is just a very small sample that can pick out if everything in place will affect an offlaner, say an Assassin or a Templar. More specifically, assuming that top mid laner has important mana needs to undertaker as timings increases.Concepts that go into wisping teamfights always revolve around a tank composition. Picking the meta we want are to at least mitigate shield leech early unless you have max hp in zaelus, paladin and kidd. Even if you are only one tank ADC with greater skill stack or if you donŽ2019t necessarily run tanky, you should always try to avoid aggressive mid laners. How strong you are, also depends on the main map your opponents have and the level of your presence on most ones. For sure that it needs to be built that you also have experience on an ally you can use on most main maps for early game control once cross biofus.

Defensive defensive champions:

The tank shield against crits/cooldown and Ept/HOME shenanigans shouldnŽ2019t take too long to turn off any tank supports. The Breacher Shield can also be used to overcome team stump!

Champions ADC Stats 4 5 10 7 8 16 18 7 Cooldown 2 6 pulses CC in 50 base armor Regem Crit Power 14120 Resist AP at max level, 430 CCD Payroll 930 Deviate AP at max level 10 Cyclone 900 Ranged Poisoning 50% on AP skills at max level, 1444 NET abilities 800 BKB PoI 2475 WCP Defensive champions in gold per attack, and by number of CR per dmg. hexcaprec.com/SPs_lane.jpg

Round 1: Mega Troll

EN!I172 1278 440 Gemgenius "Crazy" CrunchyTrade# Stats of huge Twisters reported Card Stats Quotes Tactical level MasterOver 9000 Adv1 125 WMPUlt eggoBlue File Page 316WPP 944

Converse to TT's build:

I know it would make easy a transition to KatainI would be wise to try to pick maniac alphas on Aklematic. Honestly, stuns are the worst in the game and are considered sign of turbulent terrain. TT's beautifully short animations on these are always going to be a meta property, when stuns on Elanna leavoured, jungling jungling and flying slow are the fishworks in The anime.

More precisely speaking, he can step up to TBA if she works like a climber on a butterfly, then on a cat,ale and back.

I asked reddit for its thoughts on laning when he released the very big curse Emptables. Ever since i wrote about his hegemony on my other news site, it seems we are back to juke with the topic of playtime or BlackStorm. Should are cheap, breaker to get back?

And so i ask, if TT does a slouch on his build like Alistar ormalfurr, i have a smart hunch that if Imp or TTeM can really tank harder or play a fragile role, after he can supposedly replace, the playmaker will pick on stryrolls or slower side up protector. Barreds on TT's build (crashless auto auto heal, he wouldn't really lose any value factor like Oriana or AK, and tanking AP and INTs) are one of the powerful tank attorrupts of all time.

Kennalden and baniac are in the niche, but bull1%s are good at gatstrings and base.

Pro coaching:

OP police Profile Blog Joined January 2010 New Zealand 38849 Posts #16 Wow!

The new sign from ABCR FriUndus member needs to de (Holdaround) as GPL support by Jim ·

IM being joined (@all in part) been mostly worked on for the better half a year.

I think it would be useful if we can shake up the challenge a bit.

Using armor totems: _______ noticing the misfamiliarity: https://www.starter.com/Aprium/Major__________Gear-Charging. Aprium might be related using schedule to charge all monsters Charge monsters using the charge Majority add names have proficiency of gold nodes in tanks and length of output power Core functionality are shown below:⟨|**endoftext**|⟩Cannot have a crack at tech only for at least 45 seconds Beware of spiders to stray us, let's play meta yourself?

or prefers an easy setting place various pros and cons for the purposes of sharing ideas with the Snet music.1 leaders can't handle that curation either, they like merge to delete the items rolled and procs into requirements.Since very few people have one formed⟨|**endoftext**|⟩

**Figure 9:** A sample generated by MDLM trained on OWT with 1,024 sampling steps. Topic shifts indicate abrupt, unrelated subject changes within a single passage. Incoherent sequences indicate spans where sentence structure collapses into meaningless token concatenation.

⟨|**endoftext**|⟩ off people for the first time at that point, somehow do not prolonging your woes. Secondly, it is never easy to know the culture's motives. It is also very easy to get a firm sense of what you should offer anyone if you do not know what their motives are. Sure, you don't always get the feeling that being wrong is the answer to the issue. But, it will work against you. You understand what they were and they have answered the question. Probably the most widely used saying is that it was 'after those misery and misery' BS and it is still a bad choice. There is only a single good option. Who is to target if they go so bad over, why go to war with people in black clothing.

Some of the approaches that harm the environment are incredible, but a deeper level, it is not true without the success of anything useful afterwards. But first, it helps to consider what our choices actually are. It is important to know who is in charge, how they are responding to reality.

Have you ever got a transition presentation?

In case all you are studied making this question already, I can see myself suggesting you bring someone or woman to the business club. It needs to be quite informed and clear, then there is so much work ahead that it is fully satisfied just the agenda of bureaucrats in that evil Pigguonious tibboleth. Can you relate to the students at all as almost everyone else? Certainly making a decision consists of nothing more than time. That just allows you to be reassured that you never know them better. Not aware of their meaning and abilities. Not letting things go before it starts all over in some other place too

That's a mistake. I imagine bringing a person away would end the conversation. You just can't do that. Give in your data and give yourself a platform to assess what means before you decide to mislead. Once you do, the comparator's problem for at least two months is the use of the ever changing strategy as an excuse for new situations. Once you view them as voluntary behaviour, this only grosses them out as hindrance.

Shining the Spear

There wasn't enough time to consider another question of bearings, these are ideas, opposed to necessarily things that matter. What that is opposing about them is that they totally matter. This is why many things are at the gateways of managers and co-workers and banking and government workers. Considered or non-working ideas can exist. They may be incorrect, but they are the product of perceptions of rationality and sincerity. No matter if they are wedged together and land somewhere between good guys because you could get op-room in exchange for playing your part, you can still get it. Managers trying to protect the idea usually resign to someone who opposes it. Do they do good for them than those who do? Or do they actually on better chances than those who oppose they ideas.

This was people. If anything, this is a stranger problem with people. Good reason is all there is to have. But, my question? For those, though, who fails to place an abhorrent expectation in others, then he rejects our presumptions. He is born of empathy and empathy is not something without his depth understanding however.

Suppose have a kid who thinks our culture must be equal, that is, because entrepreneurs are liberation activists so whoever controls the culture, in order to be less oppressive, will go into striving for another form. Muscle is not a thing that lead to power. It can make democracy. It can make a person who is different from people. Invisible people can grow power. People have the muscle to exercise that power. There is an inner strength. People can be a potent force against people that can't consciously attain strength. Person won. Some people. However, it is not something we have to choose. This is the opportunity. The opportunity is good. Yet it is easy to see it, like someone being unable to solve a puzzle.

That is two reasons why you needs to offer the small boy that opportunity very much. It is only a distraction and to have to think wouldn't use a distraction against oneself. Failure to think is the reason we take autonomous things away. You cannot have the will to live, freedom to do what we choose or make this. Even so we're tough at it. We haven't got the resources for your mind or your quest.

You can't. In order to be successful anywhere from now, you first must possess the power that can be been successful through success both personally and professionally. Even in the first few weeks, there are times when there is a strong force in your life who comes with as many options or goals as you are comfortable with. I know that you are always jealous when he introduces your well-earned stuff to friends or⟨|**endoftext**|⟩

**Figure 10:** A sample generated by UDLM trained on OWT with 1,024 sampling steps. Topic shifts indicate abrupt subject changes. Unlike MDLM, UDLM produces grammatically plausible sentences, but the passage still lacks topical coherence across paragraphs.

⟨|**endoftext**|⟩ There are charges of battery and assault against Smith and is expected to be arrested at 7pm Mass. Thursday.Those who are on set should contact PlanTheBlossom.com at 800-707-5574.⟨|**endoftext**|⟩As the Supreme Court is expected to rule next week over the Foreign Intelligence Surveillance Court's surveillance program to dig up the phone records of citizens and the Internet-based services they use without warrants, the Obama administration says it's close to the case, due to its broad purview.

"The FISA Court has ruled, as it always has, broad executive searches, which violate First Amendment constitutional claims and rely on just a show of cause," said Malcolm Poena, senior counsel for intelligence services at the White House.

"I urge people to reconsider the decision and pursued more vigorously in the FISC Court, and perhaps of greater importance, in Congress."

For nearly two years, the appellate court has ruled that intelligence officials are not allowed to gather phone records about Americans or foreign citizens without a court or employer request, but also without showing-cause warrants, and that the government can't ask for private stored information only if critical telecommunications infrastructure "contains a threat to the public."

"It is not clear why the administration should use DoMet before curtailing companies that honor terms for the surveillance against foreign targets," its predecessor, Marcy Wheeler of San Edward, said in a statement last week.

Dennis J. Romero, founder founder of Public Knowledge, said that the legality of such requests is an ongoing issue because of a different interpretation of the law, and that his organization is trying to draw distinctions between firms culpable and not culpable.

"A service to consumers is not always that of several firms," he said. "It is our call that the president lift orders requiring that. Likewise, is encouraged by orders enacted by previous administrations to reconsider this interpretation and to seek the full implications of the decision and to bring the case to the Supreme Court."

Contact us at editors@time.com.⟨|**endoftext**|⟩Our Planet's Eclcl now joins the fragmented bluster of the Orion cluster.

Scientists've mapped Earth's great primate cluster that once populated it throwing victory to decades of thinking that ancient reptiles and gibes were born.

The Eclcl originally formed by series of young animals losing their home, dropping off eggs on dying moors between North's Centaurus constellation and South's Adu constellation.

The more than 280 million-year-old (62.5 billion years ago) reptiles set out on the austere journeys to a new location, the 'eureka' as it is now known.

Eventually they realised the shipshape was no longer safe, so they abandoned the colonies.

"We have always wanted to get a picture of the last nursery location and whether it flowed into another," said co-author Calvei Richmond of the Atriemia de Investigadamento e Cholsa de Brazil.

"We are glad to see our map does seem to overturn longstanding fossil record and traditional hypotheses."

Richmond has spent a year probing the Ecl via three satellites on the ground but a detailed map showing the cluster's history has been elusive.

The paper map uses observations from Brazil's MARS Digital Surveys EPOS Landgrab Connector, which has been taking off from around the globe.

The study appears in the high order journal 'Nature'.

Reference: 2012-03-10 DOI: 10.1688/gpaid.imob.2012.10⟨|**endoftext**|⟩Jamout Bling!

Jamout Bling is one of the most popular, month monthly OrganConner MO. In the live group-match MOBA, players will have to compete against a variety of "smoothingles" to attempt prizes. At least have us right, Jamout Bling is solid fun with a great opportunity to make a name for itself.

What is Jamout Bling?

Jamout Bling is a small community that operates primarily by hosting prepaid prize pools of approaching 3,000, and the team made it easy for you to have fun with other Blayers. Entry fee is $300, and get ready!

Why we jam our house with the community!

Jamout Bling has run for about six months by a group of folks living within a block of Willow Creek! We have weekly jams on the spot and bantering with local Jamaican snotty bands.

When's the Jam?

Until A Friday at 10am!

As much as you want to have fun, come out! It's a very small crowd, bring your critters too, and we'll be hosting a dartboard, so just put your controller on if you have it on.⟨|**endoftext**|⟩

**Figure 11:** A sample generated by the LDDM-M trained on the OWT dataset, using 1,024 sampling steps.

⟨|**endoftext**|⟩One of the most enduring stories that will eventually lead us to this point is the "Oil War." Between 1992 and early 2000, the United States proved to be China's largest central defense partner, having a currency turnover of over $100 billion. Astonishing America's growing closer to Iran and Russia, China to turn away from Iran and relinquish its oil reserves in the Middle East. It started blocking deals with the United States. China feared that if it held Americans ransom with the oil, Middle Eastern regimes would move away from the United States as an indispensable partner to truly China, as well as acquiring the so-called "Green economy."

If a story went more like this, Iran might not even be surprised by this grand strategy. Instead, it decided to respond by physically seizing some of the vast oil reserves north of the border, near Saudi Arabia near Basra. That way, it tapped into steady streams of energy supplies from Russia and Iran. China shelled an Iranian test facility and checked out its production network for amylizing agent within its first thermonuclear megaton. Ultimately its sole concern was to hold on oil.

The price for all the Iranians' oil misadventures was massive American imports. Today, E.G.A.s sales between the Middle East and Pakistan rose by 29 percent between 2000 and 2015. Ten years ago, the cost of America's largest drone was under single-digit dollars. Today, America's largest Predator drone costs a whopping $32 billion. Imagine the consequences of Reagan's wars, such as the encroachment of the Soviet Union in the 1980s, and his successor's wars as well Korea, Vietnam, and especially East Vietnam. Likewise, oil alone is no longer a source of revenue but a big driver of investment to U.S. companies. It funded those countries, like the African Republic, which boasts the biggest military in Africa, where the price of honest (U.S). oil imports was about the price of oranges in 2000. Today, the U.S. trade on oil exports is close to $180 billion.

America's growing dependence on oil in the South Pacific, as well as all of Southeast Asia, need to grow even more as technology advances. "The core challenge to secure a balance to sustain growing liabilities," wrote Vern van den deWalk, a senior technology analyst at Fortune Global Trading Service, Fortune's analytical unit located in London, in a note he wrote. "Not so much dictated by technology but it will be the name of the game."

The huge challenge of cybersecurity will soon prove to be an important factor in the partnership's longevity, as well as the longer term consequences. While the president's role, as long as it has been known, has been the pursuit of "smart" solutions, what is at the center of that may soon shift as the decade moves forward. Threats like recent data breaches and the ensuing military conflict could further wind up the "cybersecurity" and "defense industry" worlds of government and defense.

Dalea McGovern, a CTO at the upstart Lockheed Martin, designs the F-35 Multi-Role Vehicle. (Reuters)

It does not take a bit of imagination to help imagine the likely scenario down the road. Early on likely, the world arrangements of security and defense would be one of plates, complete with large armed forces, an enquacious private sector, and an ability to participate in multiple major markets globally. Simultaneously, that industry would contain significant and esteemed government functionaries that tend to solve their respective problems, as execances are, the costs of arranging the two would vary.

Between 2004 and 2015, military agencies handled roughly a gazillion Pentagon data requests/day. If you look at estimates produced by the Department of Justice, though, the United States intelligence agency accounted for 31 percent of that load. In these projections, the Pentagon handled data roughly 400,000 civilian shooting incidents every day. No single network or information infrastructure could properly offload such massive streams of requests and devote such enormous amount of time and resources to gathering them all. The situation would reflect an increasingly wrangling and complex complex enterprise in our day's advanced strategic world.

Part of that potential comes from how inelastic things are now. Today, the military employs more than a half of its service members and twenty percent its contractors. In fact, a little less than one in five of the armed forces had served between 1900 and 1985. Before that, about a couple of out of five men fought in the Civil War. The game as small and timid as that could become, and it will require a deep strategic and professional transformation for the United States to play the role that it can.

The Bad

Most people don't imagine, but⟨|**endoftext**|⟩

**Figure 12:** A sample generated by the LDDM-U trained on the OWT dataset, using 1,024 sampling steps.

