# OpenReview forum: "Loopholing Discrete Diffusion: Deterministic Bypass of the Sampling Wall"
_ICLR.cc/2026/Conference — ICLR 2026 Poster_

### Official Review · Reviewer_Sjct · 2025-10-30

**Soundness:** 3
**Presentation:** 3
**Contribution:** 2
**Rating:** 6
**Confidence:** 3

**Summary:**

This paper identifies a key limitation in discrete diffusion models, which the authors term the sampling wall. This refers to the information collapse that occurs when the rich, continuous categorical distributions predicted by the model are sampled, reducing them to one-hot vectors for the next denoising step. The authors hypothesize that this loss of distributional information is a root cause of known inefficiencies in discrete diffusion, such as "idle steps" and "excessive oscillation" .To address this, the paper introduces Loopholing, a mechanism that creates a deterministic latent pathway to complement the standard stochastic sampling path. The resulting models, Loopholing Discrete Diffusion Models (LDDMs), are evaluated on language modeling and arithmetic reasoning. Experiments show that LDDMs significantly improve upon baseline models, achieving up to a 61% reduction in PPL. It also improves performance on reasoning tasks like Countdown and Game of 24.

**Strengths:**

a. The proposed Loopholing mechanism is a simple and logical solution to the stated problem. Propagating the continuous latent state $h_s$ is a direct way to preserve information, and the architectural modification is relatively minor but highly effective.

b. The authors successfully apply Loopholing to two different families of diffusion models (Masked Diffusion and Uniform Diffusion) and two different domains (open-ended language modeling and structured arithmetic reasoning ). This suggests that Loopholing is a general and widely applicable technique.

c. The analysis of Temporal KL (TKL) and Token-Prediction Entropy (TPE) in Figure 5 compellingly supports the hypothesis that Loopholing mitigates oscillations and idle steps.

d. Loopholling mechanism surpass of its discrete diffusion backbone even under the same computing budget.

**Weaknesses:**

a. While the results are excellent at this scale, it remains an open question whether these substantial relative gains will persist with much larger models.

b. The paper notes that an attempt to apply Loopholing only during fine-tuning was unsuccessful. This limits the capability to improve existing diffusion models. Is it possible to continue pre-train with an existing diffusion model with the loopholing mechanism?

c. Loopholing yields strong improvements for MDLM but marginal gains for UDLM (Table 2), except on PTB. The authors attribute this to domain shift sensitivity in UDLM perplexity, but this undermines the generality of the proposed mechanism. Are there any other metrics can be used to evaluate the performance?

**Questions:**

The paper modified the TopK decoding of MGDM to only apply to masked tokens. However, Table 4 in the appendix shows that the original MGDM decoding actually performs better for both the baseline and LDDM-G. Why does this approach that ‘conflicts with the training objective’ achieve higher performance?

---

> ### Author Response · Authors · 2025-11-22
>
> We thank the reviewer for the insightful and detailed comments. We especially appreciate your recognition of the simplicity and effectiveness of the loopholing mechanism, as well as the supporting analysis. Your comments have helped us further strengthen the paper. Our detailed responses follow.
>
> ## [W1] Whether these substantial relative gains will persist with much larger models
>
> We thank the reviewer for the insightful question regarding scalability. We agree that validating the method on larger models is crucial to demonstrate the robustness of our approach.
>
> While it would be ideal to evaluate Loopholing on industry scale models, this is unfortunately beyond our current computational budget in an academic setting. However, for our main experiments we carefully followed the model scale used in prior work on MDLM, so that our results are comparable to standard settings in the literature.
>
> To further investigate this question within our resource constraints, we trained a model that is about 2.5 times larger than our main configuration and **found that the advantage of Loopholing is preserved at this scale.** Concretely, we scaled the model size from 169M to 424M parameters (Depth: 24, Heads: 16, Hidden Dim: 1024). For a fair comparison under the rebuttal timeline, both the baseline (MDLM) and our model (LDDM-M) were trained for 100K steps.
>
> **1. Perplexity Analysis:** As shown in the table below, both MDLM and LDDM-M benefit from scaling, but LDDM-M achieves lower perplexity than MDLM across all datasets at 424M parameters.
>
> |Model|Size|OWT|LM1B|LAMBADA|PTB|AG News|Pubmed|Arxiv|Wikitext|
> |---|---|---|---|---|---|---|---|---|---|
> |MDLM|169M|27.20|80.10|55.45|116.54|89.00|51.47|44.27|45.46|
> |**MDLM**|**424M**|**22.49**|**68.55**|**47.49**|**78.67**|**64.68**|**40.98**|**36.15**|**34.90**|
> |LDDM-M|169M|26.14|80.00|49.94|102.67|79.84|47.58|40.59|39.93|
> |**LDDM-M**|**424M**|**20.20**|**64.97**|**42.52**|**72.13**|**53.82**|**36.88**|**32.89**|**31.10**|
>
> **2. Generative Perplexity (Gen PPL) Analysis:** For generation quality (measured by Gen PPL at 1024 denoising steps), the scalability of our method is even more pronounced.
>
> - **MDLM (424M):** 85.62 (improved from 108.94 at 169M)
> - **LDDM-M (424M):** **40.10** (improved from 49.13 at 169M)
>
> While the baseline improves with scale, **LDDM-M (424M) still outperforms MDLM (424M) by a factor of roughly 2.1x** (85.62 vs. 40.10). This confirms that the "loopholing" mechanism remains highly effective in preserving distributional information, leading to superior generation quality even in larger models.
>
> We will incorporate these scalability results into the revised version to further strengthen our claims.

---

> > ### Author Response · Authors · 2025-11-22
> >
> > ## [W2] Apply Loopholing to existing diffusion models via continued pre-training.
> >
> > We thank the reviewer for raising this important practical question. It directly concerns how Loopholing can be applied to existing pretrained diffusion models.
> >
> > Due to resource constraints, our initial attempt focused on a short fine-tuning run on the Countdown task, where we did not observe a clear improvement, and we therefore described this attempt as unsuccessful in the original version. To answer the reviewer’s question more directly, we have now conducted a controlled continued pre training experiment in the language modeling setting, and **this experiment shows that Loopholing can indeed be applied to existing diffusion models through continued pre-training.**
> >
> > We start from a standard MDLM checkpoint trained for 1M steps on OpenWebText. We then integrate the Loopholing mechanism by attaching zero initialized layer norm layers and continue pre-training with the LDDM-M self conditioning objective for up to 200K additional steps. The results are summarized below.
> >
> > **Perplexity analysis.** The adapted model (MDLM → LDDM-M, 200K) surpasses the original MDLM (1M) and nearly matches the performance of LDDM-M trained from scratch.
> >
> >
> > |                                          | OWT (trained) | LM1B      | LAMBADA   | PTB       | AG News   | Pubmed    | Arxiv     | wikitext  |
> > | -------------------------------------------- | ------------- | --------- | --------- | --------- | --------- | --------- | --------- | --------- |
> > | MDLM (1M step)                               | 23.05         | 66.73     | 48.36     | 86.33     | 68.62     | 41.94     | 37.52     | 36.30     |
> > | LDDM-M (1M step)                             | 21.90         | 69.53     | **44.22** | 85.80     | 62.55     | 39.74     | **34.96** | **33.27** |
> > | MDLM → LDDM-M (10K step)  | 22.39         | 65.83     | 45.70     | 83.45     | 61.70     | 40.57     | 36.55     | 33.92     |
> > | MDLM → LDDM-M (50K step)  | 21.91         | 65.34     | 45.09     | 82.20     | 60.53     | 39.22     | 35.59     | 33.35     |
> > | MDLM → LDDM-M (100K step) | 21.71         | 64.57     | 44.44     | 80.77     | 59.83     | **38.97** | 35.35     | 33.02     |
> > | MDLM → LDDM-M (200K step) | **21.50**     | **64.42** | 44.27     | **80.04** | **59.44** | 39.10     | 35.14     | 32.83     |
> >
> > **Generation Quality (Gen PPL) analysis.** For the adapted model (MDLM → LDDM-M, 200K), at 1024 sampling steps the Gen PPL drops from 108.94 to 51.51, closely approaching the scratch trained LDDM-M (49.13).
> >
> > | 169M | T=256 | T=512 | T=1024 | T=2048 |
> > | --- | --- | --- | --- | --- |
> > | MDLM (1M step) | 114.05 | 112.83 | 108.94 | 107.20 |
> > | LDDM-M (1M step) | **64.76** | **53.56** | **49.13** | **44.51** |
> > | MDLM → LDDM-M (10K step) | 91.04 | 82.45 | 78.31 | 74.65 |
> > | MDLM → LDDM-M (50K step) | 75.73 | 66.57 | 61.96 | 56.69 |
> > | MDLM → LDDM-M (100K step) | 71.01 | 60.72 | 57.20 | 51.59 |
> > | MDLM → LDDM-M (200K step) | 66.06 | 55.81 | 51.51 | 47.04 |
> >
> > These results indicate that continued pre-training allows an existing diffusion model to effectively adopt the Loopholing mechanism and recover most of the benefits observed when training LDDM-M from scratch.
> >
> > ## [W3] Why Loopholing yields strong improvements for MDLM but marginal gains for UDLM
> >
> >
> > We thank the reviewer for raising this point, and we would like to clarify that **Loopholing also brings substantial gains for UDLM when we evaluate with our main generation quality metrics** rather than only standard perplexity in Table 2.
> >
> > In addition to standard perplexity, which we approximate from the NELBO that serves as the training objective, we evaluate generation quality using two sample based metrics, Gen PPL and G-Eval. Because perplexity is tied to the training objective, it does not fully reflect how multi-step latent propagation improves the actual generation process, whereas Gen PPL and G-Eval are computed from generated samples and capture this effect more directly.
> >
> > For UDLM, Figure 4(a) shows that LDDM-U improves Gen PPL by about 2.5x over UDLM at 1024 sampling steps, and Figure 4(b) shows that G-Eval consistently prefers LDDM-U outputs. We will clarify this discussion in the revision.

---

> > > ### Author Response · Authors · 2025-11-22
> > >
> > > ## [Q1] Why original implementation of Topk Decoding of MGDM performs better than the modified version?
> > >
> > > We thank the reviewer for highlighting this interesting phenomenon, namely that TopK decoding over all tokens performs better than TopK decoding restricted to masked tokens, even though the latter is more closely aligned with the training regime.
> > >
> > > While a precise explanation remains open, we hypothesize that the original all token TopK decoding behaves as a mild remasking mechanism, in line with the intuition from ReMDM [1], where tokens sampled incorrectly in early steps can be reconsidered and replaced in later steps. However, we view this only as a tentative interpretation, and a more detailed analysis of decoding strategies is an interesting direction for future work.
> > >
> > > [1] Wang, Guanghan, et al. "Remasking discrete diffusion models with inference-time scaling." arXiv preprint arXiv:2503.00307 (2025).

---

> ### Comment · Reviewer_dHL3 · 2025-11-25
> **results are not convincing as perplexity cannot show actual generation quality**
>
> **I want to emphasize here that perplexity does NOT show actual generation quality.**
> Consider case of repeated generations, such text would get low perplexity but this is actually a low quality text. I strongly ask authors to report evaluation metrics on downstream tasks and include more stablished text quality metrics such as BERTscore, diversity metrics.

---

> ### Author Response · Authors · 2025-11-28
>
> Regarding the concern that repeated or low-diversity generations can artificially lower generative perplexity, we agree that such degenerate behaviors may lead to misleadingly low perplexity values.
>
> However, **this issue was explicitly acknowledged in our original paper at line 360–361.** Also, following the mitigation strategy of Zheng et al. (2024) [1], we use 64-bit sampling to avoid such degenerate behavior (Appendix C.1.1, line 687). In addition, alongside Gen PPL, we also report sentence entropy in Fig. 1 and Fig. 4(a), which directly demonstrates that token-level diversity remains normal and that our improvements are not due to collapsed or repetitive outputs.
> Beyond generative perplexity, **we also evaluate with G-eval using GPT-4.1 in an LLM-as-a-judge setting,** providing a quality signal aligned with human judgments.
>
> **Additional academic-scale downstream evaluations are aleady provided in Appendix D.2, and we also report diversity metrics such as Self-BLEU in Tables 13 and 14.**
>
> [1] Zheng, Kaiwen, et al. "Masked diffusion models are secretly time-agnostic masked models and exploit inaccurate categorical sampling." arXiv preprint arXiv:2409.02908 (2024).

---

### Official Review · Reviewer_2QKi · 2025-11-01

**Soundness:** 4
**Presentation:** 3
**Contribution:** 4
**Rating:** 8
**Confidence:** 4

**Summary:**

This paper shows strong improvements in generation and reasoning accuracy of discrete diffusion models by allowing each diffusion step to condition on the pre-logit hidden states of the previous step.  This avoids the problem of losing information in the discrete sampling step, despite the fact that these hidden states are not trained for this purpose.

This problem is fundamentally the same as the problem with backprop across a discrete sampling step, and the intuition for solving it is essentially the same as the intuition behind Gumbel softmax (i.e. mixing the complete predicted distribution with the sampled one).  But the authors identify the forward direction of this problem as a separate issue independent of the backward direction, and demonstrate its importance for diffusion models.  They also propose a practical architecture which implements this intuition in an efficient way, by conditioning on the pre-logit hidden states (mixing these pre-sampling embeddings with the embeddings from the sampled tokens) and not backproping error across this conditioning.

**Strengths:**

The model and its motivation are well explained.  The empirical results are strong, thorough and convincing, and include the obvious ablations.

Using diffusion for non-autoregressive generation of text is an important open problem, and this paper demonstrates a method for closing the gap in accuracy with auto-regressive models.  It also shows how iterative methods such as this can improve reasoning ability.

**Weaknesses:**

Identifying the problem of passing information forward across discrete sampling steps is not as novel as they claim, since it just the forward perspective on the well-known problems with backprop across a discrete sampling step.  But the method for addressing this intuition is novel, and it is novel to show that it applies even when the model has no backprop across this step.  It would have been interesting to see how this compares to a model which does backprop across this step, but this is mentioned as future work.

Equation (2) is confusing because it uses the same variables to sometimes refer to the entire sequence and sometimes refer to individual tokens within the sequence.  I think it would be clearer to add explicit token indexes where appropriate.

I wasn't convinced by the discussion of Idle Steps, since the graphs are averages and thus do not show whether individual paths have idle steps.

For the claim in lines 360-361, please mention which graph backs up this claim.

**Questions:**

Please explain if I have misunderstood the argument about Idle Steps and Excessive Oscillation.

---

> ### Author Response · Authors · 2025-11-22
>
> Thank you for the thoughtful and constructive review. We found your forward and backward perspective on the discrete sampling problem particularly insightful, and it offered a viewpoint we had not explicitly considered.
> We also appreciate your positive assessment of our work and the comments that helped us further strengthen the paper. Our detailed responses follow.
>
> ## [W1] Addressing the discrete sampling problem is novel, but comparing it to a model which does backprop across the steps would have been interesting but remained as a future work.
>
> We thank the reviewer for this insightful comment and we fully agree that comparing with a model that backpropagates across the discrete steps is important.
>
> As part of this work, **we have aleady run an additional experiment** in which we start from the self-conditioned model that was trained for 1M steps and then fine-tune it while allowing gradients to flow through the latent path. **The details and results are reported in Appendix D.5.**
>
> In this fine-tuning setting, enabling backpropagation across the additional steps did not lead to a consistent or substantial improvement over the self-conditioning baseline. However, this experiment only covers the fine-tuning regime, and since we observe clear gains from increasing the number of forward passes, we believe that a more systematic exploration of architectures and training schedules that exploit gradient flow across steps remains an important direction for future work.
>
>
> ## [W2] Equation (2) is confusing.
>
> We thank the reviewer for pointing out the potential ambiguity in our notation. While we originally used $z_t$ to denote a discrete variable for a single token, we agree that this notation can be confusing in the context of sequence generation where distinction between the entire sequence and individual tokens is crucial.
>
> As suggested, we will revise Equation (2) to include explicit token indices (e.g., using $z_t^{(l)}$ for the $l$-th token at step t) to clearly distinguish between sequence-level and token-level variables. We note that this explicit indexing is already consistent with our implementation details in Algorithm 2 (Lines 10-11 in the Appendix).
>
> This modification is not yet applied in the current version but we will inform you soon after editing this in the revision.
>
>
>
> ## [W3] The current visualization does not clearly show whether individual paths have idle steps.
>
> We thank the reviewer for the opportunity to clarify the concept of idle steps. As noted, averaged graphs do not directly show whether individual diffusion paths contain idle steps. In the revised version, we have added an analysis at the level of single trajectories and included additional plots that visualize idle steps along individual paths. These experimental results have been added to **Appendix D.8**.
>
> ## [W4] Figure reference for claim in lines 360-361.
>
> We thank the reviewer for helping us improve the clarity of the writing. The claim in lines 360–361 is supported by the behavior **shown in Figure 4 (a)** of the paper, and we updated the text to explicitly refer to this figure.
>
> ## [Q1] Explain about Idle Steps and Execessive Oscillation If I misunderstood.
>
> Thank you for the opportunity to clarify.
>
> **Regarding Idle Steps:** Your question about the visualization on the idle steps makes sense completely. So, we believe you have understood the Idle Steps correctly.
>
>
> **Regarding Excessive Oscillation:** While we did not find specific concerns raised in your review regarding this point, we would like to briefly clarify our definition to ensure alignment. Excessive oscillation [1] is defined as the instability in the generation process. It refers to cases where samples are not denoised consistently but instead exhibit overly noisy and highly inconsistent variance during denoising.
>
> [1] Wang, Wen, et al. "Time is a feature: Exploiting temporal dynamics in diffusion language models." arXiv preprint arXiv:2508.09138 (2025).

---

### Official Review · Reviewer_dHL3 · 2025-11-02

**Soundness:** 2
**Presentation:** 3
**Contribution:** 1
**Rating:** 2
**Confidence:** 5

**Summary:**

The paper introduces Loopholing Discrete Diffusion Models (LDDMs), which modify discrete diffusion LMs by adding a deterministic latent pathway (h_t) that is passed between denoising steps, in addition to the standard sampled token outputs. The idea is to “bypass the sampling wall,” defined as loss of distributional information once you collapse logits into a one-hot token during sampling. The paper also proposes a two-pass self-conditioning scheme to train this recurrent latent path without having to unroll time.
The authors claim:

lower perplexity than prior discrete diffusion baselines such as MDLM and UDLM on LM1B and OpenWebText,

lower “generative perplexity” (measured by GPT-2) and higher GPT-4.1 “consistency”/“naturalness” scores for unconditional generations,

and improved success rates on small arithmetic reasoning tasks like Countdown and Game of 24 when applied to MGDM.

**Strengths:**

* The paper articulates a clear failure mode of discrete diffusion LMs: once you sample a discrete token, you throw away the richer distributional state and force the next step to start from a one-hot. They call this the “sampling wall.”

* The proposed “loopholing” mechanism - passing a continuous latent h_t forward alongside the sampled token sequence — is architecturally straightforward and appealing for implementation.

* LDDM-M beats MDLM on perplexity on LM1B and OWT (e.g. ≤25.95 vs ≤27.60 on LM1B; ≤21.90 vs ≤23.05 on OWT).
* On unconditional generation, “Gen PPL” improves a lot (e.g. 49.13 vs 108.94) and GPT-4.1 gives higher fluency/consistency scores.
* On arithmetic reasoning puzzles, LDDM-G improves success rates vs MGDM (e.g. Countdown4: 86.5→94.4% at 85M params).

**Weaknesses:**

* Evaluation is narrow and relies only on perplexity metric.
The paper repeatedly claims “closing the autoregressive gap,” but this is based entirely on approximate perplexity (NELBO upper bounds) rather than true log-likelihood or downstream task performance.
There is no evaluation on actual generative tasks such as summarization, dialogue, open-ended story writing, or reasoning. Consequently, we cannot conclude that the method improves generation quality. It is important to evaluate the models on actual dowsntream performances, some of the current widely used benchmakrs for evaluation of the large scale language models are summarization tasks, MMLU, MMLU-Pro, Math-500, GSM8k, and reasoning benchmarks, and code benchmarks like Humaneval, MBPP, ...

* Limited novelty beyond standard self-conditioning.
The proposed “loopholing” mechanism - passing a continuous latent state between denoising steps trained via a two-pass stop-gradient trick — is almost identical to self-conditioning already used in diffusion and consistency models (e.g. Jabri et al., 2022; Chen et al., 2022).
The only real difference is that LDDM explicitly names and propagates a latent vector rather than the previous clean prediction. This is a minor architectural variant, not a new principle.

* The “sampling wall” is largely a renamed version of known problems.
What the paper calls the “sampling wall” (loss of information when discretizing predictions) is conceptually the same as the “idle steps” and “oscillation” problems already identified in earlier discrete diffusion work (e.g., MDLM, SSD-LM).
The paper renames these issues and overstates their novelty, without providing direct causal evidence that loopholing specifically resolves them.

* Causal explanation lacks experimental proof.
The claim that the propagated latent  “bypasses” the sampling wall is asserted, not demonstrated.
There is no ablation isolating: the effect of the latent path from the effect of self-conditioning, or the effect of feeding forward x_{theta, t} versus h_t
	​
* The observed differences could easily stem from generic stabilization due to the two-pass stop-gradient process.

* Weak baseline fairness and inherited limitations from MDLM.
LDDM inherits MDLM’s fairness issues:
Perplexity is computed as an upper bound, which is not directly comparable to autoregressive likelihoods.


* Trivial or overly simple benchmarks.
Language modeling experiments are limited to LM1B and OpenWebText at relatively small scale.
The “reasoning” benchmarks (Countdown, Game of 24) involve symbolic arithmetic and tiny vocabularies.
These tasks do not capture the complexities of realistic generation (e.g., global coherence, factual reasoning, long context).
The method is not tested on any challenging downstream generation benchmarks (summarization, QA, GSM8K, MATH, etc.).

* Efficiency and compute cost not substantiated.
The paper claims fewer idle steps and higher efficiency, but provides no wall-clock, FLOP, or token-throughput comparisons to MDLM or UDLM.
Training is actually ~30% slower due to doubled embeddings and latent propagation overhead.
There is no evidence that LDDM improves quality per unit of compute.

**Questions:**

* Novelty vs self-conditioning: How does loopholing differ from existing self-conditioning techniques beyond storing an explicit latent ?
* Please include an ablation where standard self-conditioning is applied to MDLM without your recurrent latent path, to show what the new component adds.

* Isolation of latent path effect:
Can you run controlled ablations for:
(a) no recurrent latent but with self-conditioning;
(b) recurrent latent but single-pass training;
(c) caching and reusing previous x_{\theta, t} instead of h_t?
	​This would clarify which mechanism drives the improvements.

* Task breadth:
The current benchmarks (LM1B, OWT, Countdown, Game of 24) are short and low-complexity.
Can you evaluate on actual generation tasks such as summarization (CNN/DailyMail, XSum), long-form reasoning (GSM8K, MATH), or instruction-following datasets?
Without these, it’s unclear if LDDM meaningfully improves text generation quality.

* Perplexity vs real performance:
Since the evaluation focuses on approximate NELBO perplexity and GPT-based metrics, can you show concrete qualitative examples or human evaluations demonstrating that loopholing produces better text (fluency, coherence, factuality) than MDLM or AR baselines?

* Efficiency claim:
Can you provide real compute metrics (tokens/sec, FLOPs/sample, latency for 1k tokens) to support the “higher efficiency” claim?
Does LDDM actually require fewer diffusion steps at equivalent output quality?

* Baseline fairness:
Are the MDLM and AR baselines trained with equal data, compute, and precision settings?
How sensitive are the reported improvements to these factors?

* Robustness of evaluation modifications:
For MGDM, your evaluation differs from Ye et al. (2024). Can you report results under the original evaluation scheme for transparency?

---

> ### Author Response · Authors · 2025-11-22
>
> Since the review was revised right before the soft deadline, we will first proceed with what we prepared for the previous version of the review.
>
> ## [W1-a] Evaluation relies entirely on approximate perplexity using NELBO upper bounds.
>
> We thank the reviewer for raising this point, which gives us the opportunity to clarify our evaluation metrics.
>
> **We do not rely solely on approximate  NELBO-based perplexity.** Generative Perplexity (Gen PPL) (Figs. 1 and 4(a)) measures the exact log-likelihood of generated samples under a pretrained GPT-2 Large model, providing a widely used proxy for sample quality. Additionally, we utilize G-eval, employing GPT-4.1 to score the samples in an LLM-as-a-judge setting
>
> ## [W1-b, W7, Q4] Evaluation is restricted to small-scale settings. More large-scale downstream tasks (MMLU, GSM8K, etc.) are required to assess generation quality.
>
>
> We thank the reviewer for raising the concern about benchmark complexity.
>
> **The LM1B and OpenWebText datasets, along with Perplexity and Gen PPL metrics, are standard benchmarks in the discrete diffusion literature, utilized by numerous works** (e.g., MDLM , MDM-prime [1], Duo [2], and Block Diffusion [3] etc.). Similarly, the Countdown task is a commonly used benchmark for reasoning (e.g., MGDM , Time is a Feature [4]). Our goal was to ensure direct comparability with this line of work. Benchmarks designed for large-scale language models (e.g., MMLU, GSM8K, HumanEval) are typically not feasible for the academic-scale discrete diffusion LMs considered in these studies.
>
> **We acknowledge the importance of downstream evaluation and have reported results in Appendix D.2.** We selected tasks (ARC, HellaSwag, LAMBADA, etc.) following protocols from academic-scale models like Mamba [5], PGM [6], and Scaling up masked diffusion models [7]. Evaluating on highly challenging benchmarks for large-scale models, such as MMLU, GSM8K, and code generation, remains an important direction for future work, primarily due to the computational costs required to scale our models to that regime. We will revise our paper to include this discussion and thank the reviewer for raising this concern.
>
> [1] Chao, Chen-Hao, et al. "Beyond masked and unmasked: Discrete diffusion models via partial masking." The Thirty-ninth Annual Conference on Neural Information Processing Systems. 2025.
>
> [2] Sahoo, Subham Sekhar, et al. "The diffusion duality." arXiv preprint arXiv:2506.10892 (2025).
>
> [3] Arriola, Marianne, et al. "Block diffusion: Interpolating between autoregressive and diffusion language models." arXiv preprint arXiv:2503.09573 (2025).
>
> [4] Wang, Wen, et al. "Time is a feature: Exploiting temporal dynamics in diffusion language models." arXiv preprint arXiv:2508.09138 (2025).
>
> [5] Gu, Albert, and Tri Dao. "Mamba: Linear-time sequence modeling with selective state spaces." First conference on language modeling. 2024.
>
> [6] Deschenaux, Justin, Lan Tran, and Caglar Gulcehre. "Partition Generative Modeling: Masked Modeling Without Masks." arXiv preprint arXiv:2505.18883 (2025).
>
> [7] Nie, Shen, et al. "Scaling up masked diffusion models on text." arXiv preprint arXiv:2410.18514 (2024).
>
> ## [W1-c, W6] Perplexity is computed as an upper bound, which is not directly comparable to autoregressive likelihoods. But paper claims “closing the autoregressive gap”
>
> We thank the reviewer for pointing out the potential fairness issue inherited from MDLM.
>
> **Basically, we do not compare discrete diffusion models and autoregressive models using the NELBO-approximated perplexity metric.** The test perplexities in Table 1 and the zero-shot perplexities in Table 2 are only used for direct comparisons among discrete diffusion models such as MDLM and UDLM, which all share the same upper bound formulation.
>
> **Whenever we compare LDDM with an autoregressive model, we instead use generative perplexity (Gen PPL) as reported in Fig. 1 and Fig. 4 (a).** As discussed in **[W1-a]**, Gen PPL is computed by evaluating generated samples under a pretrained GPT-2 Large model, so it corresponds to the exact log-likelihood of these samples under that reference model and allows a fair comparison between diffusion and autoregressive models.
>
>
> Regarding the phrase “closing the autoregressive gap,” we agree that, as originally written, it may sound broader than what is supported by our evaluation. Our intent was to describe the gap only in terms of likelihood and Gen PPL-based metrics standard in the discrete diffusion literature, and we will revise the paper to make clear that our claims apply only to the metrics and settings evaluated in this work.

---

> > ### Author Response · Authors · 2025-11-22
> >
> > ## [W2, Q1] Limited novelty beyond standard self-conditioning except latent propagation.
> >
> > We thank the reviewer for this critical question, which allows us to clarify our core contribution.
> >
> > We respectfully disagree with the characterization of Loopholing as a minor architectural variant. To our knowledge, no prior discrete diffusion model employs a dual pathway (stochastic tokens + deterministic prediction) during generation. **Our novelty lies in identifying and solving fundamental limitations specific to discrete diffusion that standard self-conditioning cannot address:**
> >
> > 1. **Information-Poor Binary States:** Unlike continuous diffusion, where inputs are rich, noisy blends, discrete inputs (tokens or [MASK]) provide very weak signals.
> > 2. **The Sampling Wall:** Categorical sampling collapses rich distributional predictions into one-hot vectors. Standard self-conditioning (e.g., Chen et al., 2022) propagates the predicted output (post-collapse), meaning that critical uncertainty and likelihood information is already lost during sampling.
> >
> > **Core Contribution:** Loopholing propagates the internal latent $h_t$ (pre-collapse) rather than the output, enriching the sparse binary input with continuous context and bypassing the Sampling Wall by preserving information before discretization. Self-conditioning is used only as a training technique to avoid prohibitive backpropagation through time, not as the main novelty. As shown in our response to **[Q2]**, standard self-conditioning (propagating a sampled clean estimate) yields significantly lower performance than propagating continuous latent, confirming that the continuous latent architecture is responsible for the improvements (up to 61% Gen PPL reduction).
> >
> > ## [W3-a] The “sampling wall” is largely a renamed version of known problems. It is conceptually the same as the “idle steps” and “oscillation” problems already identified in earlier discrete diffusion work (e.g., MDLM, SSD-LM).
> >
> >
> > We thank the reviewer for this insightful comment, which allows us to clarify our conceptual contribution more precisely.
> >
> > We agree that "idle steps" and "excessive oscillation" are known inefficiencies in discrete diffusion, and we **explicitly cite this prior work** throughout our paper .
> >
> > However, we respectfully disagree that the "Sampling Wall" in discrete diffusion is a "renaming" of these issues. Instead, it is the **identification of the *common underlying cause*** that leads to these previously observed **symptoms**.
> >
> > **Prior Work Observed Symptoms, Not the Cause.**
> >
> > - The reviewer correctly points to **MDLM (Sahoo et al., 2024)**. This work indeed observes the symptom of "idle steps" in Section 4.1 (i.e., when no new tokens are unmasked). **However, it does not diagnose this as a fundamental problem of information loss.** Instead, it treats this behavior primarily as an **optimization opportunity** for faster inference via caching.
> > - SSD-LM (Han et al., 2023) investigates various logit projection strategies (e.g., Greedy, Sampling, Multi-hot) to perform discrete sampling. **However, their work frames this as a performance trade-off to be tuned** (see their Fig. 4), rather than diagnosing the sampling step itself as a fundamental bottleneck of information loss.
> >
> > Thus, prior work clearly documented the *symptoms* but did not, to the best of our knowledge, diagnose the underlying cause.
> >
> > ## [W3-b] the paper don’t provide direct causal evidence that loopholing specifically resolves idle steps and oscillations.
> >
> > We thank the reviewer for raising this important point, which concerns whether Loopholing actually addresses these problems.
> >
> > **We already show that resolving the Sampling Wall mitigates these symptoms.** As detailed in our ablation study (Section 6.3):
> >
> > - In Figure 5(b), we define Temporal KL divergence (TKL) to quantify generation progress. LDDMs exhibit significantly higher TKL during the exploration phase compared to baselines. This proves that propagating the latent state ($h_t$) prevents the model from stagnating (idling) by allowing continuous updates even when the discrete token remains unchanged.
> > - In Figure 5\(c\), we measure Token-Prediction Entropy. LDDMs maintain consistently lower entropy throughout the process. This confirms that preserving distributional context allows the model to be more confident and stable, directly resolving the excessive oscillation caused by “predicting from scratch”.
> >
> > These experiments show that our Loopholing mechanism can mitigate the idle step and oscillation problems.

---

> > > ### Author Response · Authors · 2025-11-22
> > >
> > > ## [W4-a, Q3-a]  Bypassing the sampling wall is asserted, not demonstrated. No ablation isolates the effect of the latent path from that of self-conditioning.
> > >
> > > We appreciate the reviewer’s request for clearer causal evidence behind the effect of latent propagation.
> > >
> > > **We aleady conducted a dedicated ablation study.** In Section 6.3, Fig. 5(a) shows an experiment that varies only how far the latent is propagated.
> > >
> > > Specifically, we generate sample with 1024 denoising steps and introduce a reset interval $k$. Every $k$ steps, the contextual latent $h_t$ is **reset to the self-conditioned latent** instead of being propagated. when $k=1$, effectively **no latent propagation** and always use the current self-conditioned latent as requested at each step. when $k=1024$, full propagation across the whole generation process.
> > >
> > > As shown in Fig. 5 (a), $k$ increases, generative perplexity improves monotonically:
> > >
> > > - **LDDM-M:** 99.23 to 49.13 (from $k=1$ to $k=1024$)
> > > - **LDDM-U:** 63.07 to 27.84 (from $k=1$ to $k=1024$)
> > >
> > > Thus, **propagating the latent state consistently improves generation quality, which is exactly the causal effect we attribute to the latent path that “bypasses” the sampling wall.**
> > >
> > > ## [W4-b, Q3-b,c] Can you run controlled ablations for (b) recurrent latent but single-pass training; \(c\) caching and reusing previous $x_{\theta, t}$ instead of $h_t$.
> > >
> > > We thank the reviewer for suggesting these ablations, which helped us clarify the role of each component.
> > >
> > > **For (b) (Recurrent latent without training).** We confirm that **self-conditioning training is necessary for the latent path to be usable in practice.** We started from a 1M-step pretrained MDLM on OpenWebText and added the recurrent latent path only at inference by simply adding the latent embedding from the previous step. Without retraining the model to consume this path, generation at 1024 steps yields a Gen PPL on the order of $5 \times 10^5$, which indicates that the model fails to produce meaningful samples.
> > >
> > > **For \(c\) (Propagating logits $x_{\theta,t}$ instead of $h_t$).** We confirm that both **latent and logit propagation outperform the baseline MGDM**. This directly supports the claim that **bypassing the sampling wall by propagating pre-sampling information (latent or distribution) is causally beneficial**. Since language modeling has a large vocabulary size ($\approx 50{,}000$), propagating full distribution information is computationally expensive. Therefore, we conducted this experiment on the Countdown task, which has a smaller vocabulary size ($\approx 31$), using a 6M-parameter model. We trained three models: **MGDM (baseline, no loopholing)**, **LDDM-G (latent)** which propagates the latent state, and **LDDM-G (logit)**, which propagates $x_{\theta,t}$ (logits).
> > >
> > > The results are shown below:
> > >
> > > | **6M parameter** | **Countdown 4** | **Game of 24** |
> > > | --- | --- | --- |
> > > | MGDM | 45 | 12 |
> > > | LDDM-G (latent) | 56.3 | 28 |
> > > | LDDM-G (logit) | 53.4 | 20 |
> > >
> > > The latent path consistently outperforms the logit path in this setting. We hypothesize that this is because the latent has a more abstract representation than logits, and therefore can encode richer contextual information.
> > >
> > > ## [Q2] Please include an ablation where standard self-conditioning that propagates sampled clean prediction is applied to MDLM without your recurrent latent path.
> > >
> > > Thank you for this careful question about the contribution of the recurrent latent path beyond standard self-conditioning.
> > >
> > > We implemented a variant closer to standard self-conditioning (e.g., Chen et al., 2022) where the **sampled clean estimate $\tilde{x}_0$** (derived from $x_{\theta,t}$) is propagated to the next step instead of the latent state $h_t$.
> > >
> > > Due to limited compute and rebuttal time, we trained this variant for 200K steps and compared it to MDLM and LDDM-M trained for the same number of steps. We then measured Gen PPL at 1024 denoising steps using a pretrained GPT-2 Large model, with the results shown below.
> > >
> > > | 169M parameters | T=1024 |
> > > | --- | --- |
> > > | MDLM (200K) | 103.52 |
> > > | LDDM-M (Latent, 200K) | **55.04** |
> > > | LDDM-M( Sample, 200K) | 93.71 |
> > >
> > > While sample-based propagation offers a slight improvement over MDLM, it lags significantly behind latent propagation. This confirms that **preserving pre-sampling information via a continuous latent path is crucial**, whereas recycling **post-sampling predictions** (standard self-conditioning) is **insufficient to overcome the sampling wall**.

---

> > > > ### Author Response · Authors · 2025-11-22
> > > >
> > > > ## [W5] The observed differences could easily stem from generic stabilization due to the two-pass stop-gradient process.
> > > >
> > > > We thank the reviewer for this helpful comment, which helps us clarify which component of our method is responsible for the observed gains.
> > > >
> > > > **Gen PPL shows substantial improvements even without the two-pass stop-gradient process.** As discussed in **[W1-a]**, Gen PPL is computed on actual samples generated by the model. At inference, these samples are produced with latent propagation rather than with the two-pass stop-gradient procedure used during training. Under this sampling process, LDDM achieves substantially lower Gen PPL than MDLM, as shown in Fig. 1 and Fig. 4. In addition, when we evaluate the same generated samples with GPT-4.1 using the G-Eval protocol, LDDM also obtains consistently higher scores than the baselines.
> > > >
> > > > Also, as discussed in **[W4-a, Q3-a]**, generating samples with a simple two-pass scheme without propagating the latent state across steps leads to much worse Gen PPL than generating with latent propagation. Taken together, **these observations indicate that the improvements are not well explained by generic stabilization from the two-pass procedure alone but instead reflect a genuine gain in generation quality due to latent propagation.**
> > > >
> > > > ## [W8, Q6] Efficiency and compute cost not substantiated. The paper claims fewer idle steps and higher efficiency, but provides no wall-clock, FLOP, or token-throughput comparisons.
> > > >
> > > > We appreciate the reviewer’s concern and clarify that our efficiency claims refer to **efficiency relative to the unrolled ablation** and **inference-time sampling efficiency**, not wall-clock or FLOP reductions during training.
> > > >
> > > > **1. Training efficiency**
> > > >
> > > > Our claim in Section 3.2 concerns **computationally-efficient training of loopholing models compared to the alternative of full trajectory unrolling**. We agree that training incurs ~30% overhead due to latent propagation. But without self-conditioning, training would require multi-step unrolled denoising (Appendix D.5), which is far more computationally expensive.
> > > >
> > > > To make this concrete, on 8 RTX 4090 GPUs we observe the following training times for 10,000 steps:
> > > >
> > > > |  | Hours per 10,000 steps |
> > > > | --- | --- |
> > > > | MDLM (original implementation) | 4.46 |
> > > > | LDDM with self conditioning (2 step) | 5.22 |
> > > > | 3 step unrolled training | 18.24 |
> > > > | 4 step unrolled training | 34.28 |
> > > >
> > > > Training time **scales prohibitively** with the number of unrolled steps. In this sense, the proposed self conditioned scheme provides a practical and relatively efficient way to train LDDMs, even though it introduces modest overhead relative to MDLM and UDLM.
> > > >
> > > > **2. Inference efficiency**
> > > >
> > > > **Our efficiency claim in Section 4 is about generation efficiency per denoising step.** Loopholing reduces idle steps and accelerates progress along the denoising trajectory, so that LDDMs reach higher quality at the same number of sampling steps.
> > > >
> > > > This is supported by Fig. 1 and Fig. 4(a): at 128 steps, LDDM achieves better Gen PPL than MDLM and UDLM even when those models use more steps, and across all step counts LDDM obtains substantially better Gen PPL with identical compute per step.
> > > >
> > > > Thus, while training overhead exists, the mechanism yields significant inference time quality per compute gains under fixed step budgets. We will revise the paper to make this distinction between training and inference efficiency explicit and to avoid potential misunderstandings.

---

> ### Author Response · Authors · 2025-11-22
>
> ## [Q5] Can you show concrete qualitative examples or human evaluations demonstrating that loopholing produces better text (fluency, coherence, factuality) than MDLM or AR baselines?
>
> We appreciate the reviewer’s request for concrete qualitative evidence. While perplexity and GPT-based metrics provide a quantitative overview, we agree that inspecting actual samples is crucial to understanding the practical benefits of Loopholing.
>
> First, we examine a typical failure case from the **MDLM (Baseline)**. As shown below, while the model generates locally valid syntax, it suffers from severe semantic drift, rapidly losing the initial context:
>
> - "If Secretary of State was a time manipulator caught off guard, he is a U.S. ambassador from the European Union and Kiev diplomat. Obama wore a sensible beard and a much-sanctioned Rolex watch. He got to sit in a deck chair on the beachfront and smell a Aztec bundle of fried rice the Kremlin wanted Ukraine to eat."
>
>
> As seen in this example, the generation suffers from a lack of consistency. The model fails to maintain the **central subject, generating a sequence of nonsensical and disjointed statements.**
>
> In contrast, for **LDDM-M (Ours)**, we selected a sample that shares a similar thematic starting point, referencing a "Secretary of State." (Note: Since our model is trained on OpenWebText, generated samples often reflect the political news content typical of the dataset.) This allows us to demonstrate how the model handles similar contexts:
>
> - "Former US Secretary of State Hillary Clinton said Monday she would 'wait out the time' for political reform in Saudi Arabia if she wins for president in the 2016 election – a call that is opposed by Western conservatives."
>
>
> Unlike the baseline, LDDM-M successfully maintains the context throughout the sequence. It correctly **connects the presidential election with political reform, demonstrating that our Loopholing mechanism enables consistent and coherent generation.**
>
> ## [Q7] Baseline fairness: Are the MDLM and AR baselines trained with equal data, compute, and precision settings? How sensitive are the reported improvements to these factors?
>
> We thank the reviewer for the question. All diffusion models (MDLM, UDLM, and LDDM variants) and Autoregressive model were trained under **identical settings**: the same dataset (OpenWebText), batch size, optimizer configuration, and 1M training steps. For sampling, following Zheng et al. (2025), we use **float64 precision**, which is known to be critical for stable discrete diffusion sampling. lower precision (float32) can reduce token diversity which leads to deceptively low generative perpleixty.
>
> Regarding compute fairness, LDDM performs **two forward passes per training step** due to self-conditioning. To ensure that MDLM is not disadvantaged by receiving fewer total forward passes, we additionally trained and evaluated an MDLM baseline for **2M steps**, **matching the total number of forward computations in Appendix D.6.**  The evaluation result trends remain essentially unchanged, indicating that the observed improvements are **not attributable to unequal compute budgets**.
> These results suggest that LDDM’s gains are robust and not sensitive to the specific training allocation or precision settings beyond the standard float64 requirement shared by all discrete diffusion models. We will clarify this experimental fairness more explicitly in the revision.
>
> ## [Q8] For MGDM, your evaluation differs from Ye et al. (2024). Can you report results under the original evaluation scheme for transparency?
>
> We appreciate the reviewer’s request for transparency. As discussed in the paper, our primary evaluation adopts a stricter protocol than Ye et al. (2024) by explicitly filtering out “hallucinated” solutions that use numbers not present in the input or not constructed from intermediate computations.
>
> We additionally ran LDDM-G and MGDM **under the original evaluation scheme** of Ye et al. (2024) and used the **original TopK decoding scheme**, which recalculates probabilities over all tokens (masked and unmasked) at each step, without filtering out such invalid expressions. **The relative performance trend remains unchanged**, and LDDM-G continues to outperform MGDM across tasks. Concretely, under the original evaluation scheme we obtain the following results:
>
> |  | param | CD 4 | Game of 24 | CD 5 |
> | --- | --- | --- | --- | --- |
> | MGDM (retrained) | 6M | 51.5 | 32 | 22 |
> |  | 85M | 88.1 | 62 | 52 |
> | LDDM-G (Ours) | 6M | 60.6 | 45 | 23.6 |
> |  | 85M | 95.3 | 82 | 55.2 |
>
> We will add these results to the revised manuscript for transparency and reproducibility.

---

### Author Response · Authors · 2025-11-23
**To All Reviewers: A Serious Concern about Reviewer dHL3**

Dear Reviewers (and AC),

Thank you very much for your time and service in reviewing our paper.

We would like to respectfully bring to your attention a **serious concern** regarding the review from **reviewer dHL3**, as certain aspects of the process raise questions about the fairness and credibility of the evaluation. Our intention is not to make accusations, but to ensure that all reviewers are aware of the **factual circumstances** surrounding this review.

We summarize the relevant facts as follows:

1. An established AI-generated review detector classified reviewer dHL3’s original review as “**fully AI-generated**,” while the other two reviews for our submission were classified as “**fully human-generated**.” (https://iclr.pangram.com/reviews?submission_number=18113)

2. A few days ago, the ICLR 2026 official blog published a notice acknowledging concerns about low-quality or AI-generated reviews, and stating that papers associated with reviewers producing such reviews may be subject to rejection. (https://blog.iclr.cc/2025/11/19/iclr-2026-response-to-llm-generated-papers-and-reviews/)

3. Only a few hours after this official post appeared, reviewer dHL3 began to make **substantial changes** to the original review. Such major revisions at this timing are very uncommon, especially since authors typically have already begun preparing their rebuttals based on the initial reviews. In our case, these changes occurred only less than a day before the soft rebuttal deadline, at which point our rebuttal to the original review had already been finalized.

4. During this modification, the reviewer also changed the score from **2 to 0**. Such an unusually low score—combined with the significant gap relative to the other reviewers’ scores (**6 and 8**)—amplifies our concern about the consistency and reliability of the evaluation.

5. Importantly, all of these actions occurred **before** we submitted any rebuttal or clarification. Thus, they cannot be interpreted as a response to our arguments or feedback.

We are sharing these facts with full respect for the review process and the reviewers involved. Our goal is simply to ensure full transparency so that all reviewers can make informed assessments during the discussion phase.

Thank you again for your time and thoughtful consideration.

---

> ### Comment · Reviewer_dHL3 · 2025-11-25
> **Response to Authors’ Concern About My Review**
>
> I would like to clearly state that my review was not AI-generated. I wrote the review myself after carefully reading the submission, cross-checking the methodology, and examining the provided samples and evaluations. Any automated classifier labeling my review as "AI-generated" is incorrect. These systems are widely known to misclassify technical writing, and this appears to be the case here.
> Also rebuttal ends on Dec 03 '25 10:00 PM CET, I am so confused why you are mentioning rebuttal deadline is ended several days ago. This is definitely not true.
>
> To avoid any ambiguity:
> My assessment and my final score are entirely grounded in the technical content of the paper.
> The issues I identified are substantial, fundamental, and in my view cannot be resolved with minor revisions. Below I restate the key technical problems that led me to a strong reject recommendation:
>
> **Severe weaknesses in evaluation methodology.**
> The paper relies nearly exclusively on perplexity, which is not a reliable indicator of generation quality or coherence of generated text. As I emphasized in my review, perplexity can be artificially low even when text quality is extremely poor (e.g., repeated patterns). Coherence, diversity are not measured.
> I explicitly listed multiple relevant metrics the authors should have used (D-1/D-4, BERTScore, LLM-as-judge coherence evaluation, etc.), none of which were used in the paper.
>
> **Very poor quality of generated samples in the appendix.**
> The samples from both baseline and proposed method show major coherence failures. This raises concerns about whether the implementation is correct, whether baselines were tuned properly, or whether the model actually improves anything in practice. These issues are concrete and observable, not subjective.
>
> **Lack of evaluation on meaningful downstream tasks.**
> The benchmarks used (LM1B, OWT, small arithmetic puzzles) are extremely limited and do not demonstrate real improvements in generation quality. To claim better text generation, one must evaluate on summarization, reasoning, long-form generation, code, or other standard downstream tasks.
> Without these, the paper does not support its main claims.
>
> **No comparison to strong SOTA autoregressive models.**
> It is impossible to assess the contribution when the proposed approach is not compared to established, competitive LM baselines.
>
> **Lack of novelty relative to self-conditioning.**
> As I noted, the proposed “loopholing” mechanism is almost identical to self-conditioning, which already exists in diffusion models. The paper does not clearly articulate what is new and does not provide ablations isolating the effect of the new component.
>
> **Unsupported efficiency claims.**
> The paper claims improved efficiency without reporting wall-clock speed, FLOPs, or token throughput which are standard requirements for such a claim.
>
> These are not minor or cosmetic issues. They affect the core validity, novelty, and scientific contribution of the work. Based on these concerns, I concluded that the paper requires a major rewrite with substantially stronger evaluations, clearer novelty, improved experimental rigor, and coherent sample quality.
>
> Therefore, my score is entirely consistent with the problems I identified and reflects the current quality of the submission, not any external factor.
>
> I stand by the content of my review.

---

> > ### Author Response · Authors · 2025-11-28
> >
> > We respectfully disagree with most, if not all, of the comments above. Many of the reviewer’s claims are factually incorrect, pertain to points already addressed in the original paper, or stem from a lack of understanding or familiarity of the research area. Although the reviewer selected the highest confidence level (5), the comments suggest that while the reviewer may have experience with general large language models, **the reviewer is highly unfamiliar with the established standard practices specific to this area—Discrete Diffusion Language Models.**
> >
> > **Although we were open to a constructive discussion, the reviewer assigned a score of 0 with confidence level 5 even before we respond anything, an extremly unusual behavior.** The rebuttal process was further complicated by the reviewer making substantial changes to their review immediately before the soft rebuttal deadline. While the reviewer claims not to have been aware of this deadline, we verified that the soft deadline was clearly communicated to all reviewers via email.
> >
> > Below, we provide detailed responses to each of the reviewer’s points.

---

> > > ### Author Response · Authors · 2025-11-28
> > >
> > > ## Severe weaknesses in evaluation methodology.
> > >
> > > ### (1/2) Evaluation is weak because it heavily relies on perplexity, which is not reliable.
> > >
> > > **This claim is clearly incorrect. Perplexity is a well-established and widely accepted standard metric in the field of Discrete Diffusion Language Models (DDLM).** The most foundational and influential recent works—MDLM [1] (NeurIPS 2024, 312 citations), SEDD [2] (ICML 2024, 272 citations), D3PM [3] (NeurIPS 2021, 1525 citations)—as well as numerous follow-up studies [4–8], all adopt the same perplexity-based metrics as their primary evaluation criteria. In this research line, using these metrics is a de facto community standard and is essential for ensuring fair, apples-to-apples comparison across methods. Given this context, it is somewhat surprising that the reviewer characterizes the use of these standard metrics as a major weakness.
> > >
> > > **We have already clearly demonstrated in the original paper, not in our rebuttal, that the reliability issue of Gen PPL does not happen in our case.** Like most evaluation metrics, the perplexity metric also has some limitations despite being a standard metric. The degenerate case the reviewer mentioned is one such limitation that *may occur or may not*. We were fully aware of this issue. Therefore, in our original paper, we have already clearly addressed that this degeneration is not happening in our case by providing the sentence entropy (Fig. 1, Fig. 4(a) and line 360-361). In addition, as [9] identified why this issue occurs and provided a solution, we have applied the solution in our evaluation (in line 687-689). Because these elements are already explained in the paper clearly, we are unsure why the reviewer assumes that we do not account for this issue.
> > >
> > > ### (2/2) Coherence, diversity are not measured. (LLM-as-judge, BERTScore etc.)
> > >
> > > **This statement is also incorrect because these are already in the paper.** **Coherence.** in Fig. 4(b), our paper already includes exactly what the revivewer asks, an LLM-as-judge evaluation, as a main metric (G-eval) [10]. G-eval measures coherence and text naturalness using llm-as-judge. **Diversity.** In Table 13 and 14 in the Appendix D.7, we already reported a diversity measure, self-BLEU [11]. If the reviewer overlooked this due to its placement, we are happy to move it to the main paper for visibility.
> > >
> > > [1] Sahoo, Subham, et al. "Simple and effective masked diffusion language models." Advances in Neural Information Processing Systems 37 (2024): 130136-130184.
> > >
> > > [2] Lou, Aaron, Chenlin Meng, and Stefano Ermon. "Discrete diffusion modeling by estimating the ratios of the data distribution." _arXiv preprint arXiv:2310.16834_ (2023).
> > >
> > > [3] Austin, Jacob, et al. "Structured denoising diffusion models in discrete state-spaces." _Advances in neural information processing systems_ 34 (2021): 17981-17993.
> > >
> > > [4] Chao, Chen-Hao, et al. "Beyond masked and unmasked: Discrete diffusion models via partial masking." The Thirty-ninth Annual Conference on Neural Information Processing Systems. 2025.
> > >
> > > [5] Sahoo, Subham Sekhar, et al. "The diffusion duality." arXiv preprint arXiv:2506.10892 (2025).
> > >
> > > [6] Arriola, Marianne, et al. "Block diffusion: Interpolating between autoregressive and diffusion language models." arXiv preprint arXiv:2503.09573 (2025).
> > >
> > > [7] Rout, Litu, Constantine Caramanis, and Sanjay Shakkottai. "Anchored Diffusion Language Model." arXiv preprint arXiv:2505.18456 (2025).
> > >
> > > [8] Xie, Tianyu, et al. "Variational Autoencoding Discrete Diffusion with Enhanced Dimensional Correlations Modeling." arXiv preprint arXiv:2505.17384 (2025).
> > >
> > > [9] Zheng, Kaiwen, et al. "Masked diffusion models are secretly time-agnostic masked models and exploit inaccurate categorical sampling." _arXiv preprint arXiv:2409.02908_ (2024).
> > >
> > > [10] Liu, Yang, et al. "G-eval: NLG evaluation using gpt-4 with better human alignment." arXiv preprint arXiv:2303.16634 (2023).
> > >
> > > [11] Zhu, Yaoming, et al. "Texygen: A benchmarking platform for text generation models." The 41st international ACM SIGIR conference on research & development in information retrieval. 2018.

---

> > > > ### Author Response · Authors · 2025-11-28
> > > >
> > > > ## The poor quality of the samples in the appendix raises concerns about implementation correctness or the proposed model's ability.
> > > >
> > > > ### (1/2) The sample quality is poor.
> > > > **While it is true that the samples are not comparable to those from state-of-the-art, industry-scale LLMs, this is entirely expected.** The goal of our paper is not to compete with such models, but to improve generation quality relative to prior DDLMs under the same model scale and dataset, ensuring a fair, apples-to-apples comparison. Following established practice in this research line, we use the standard configuration adopted in previous works [1,3]: a 169M-parameter model trained on OpenWebText.
> > > >
> > > > **It is important to note that the sample quality of our model is consistent with that of standard DDLM models**, including the original MDLM (not trained by us). For clarity, we provide generated samples from this original MDLM, which exhibit similarly limited sample quality.
> > > >
> > > > **Rather than relying on subjective impressions as the reviewer does, our evaluation follows principled quantitative metrics.** Both Gen PPL and G-Eval (GPT-4.1) consistently show that our model produces more natural and coherent samples than the baselines.
> > > >
> > > > [Link for Generated Samples](https://anonymous.4open.science/r/samples-39E4/).
> > > >
> > > > ### (2/2) Due to the sample quality, the implementation may be not correct and the model's ability is suspicious.
> > > > **We respectfully disagree. We have concrete evidence that our implementation is correct.** First, the Gen PPL of the MDLM models reproduced in our work closely matches the Gen PPL reported in prior studies—for example, DUO [1] and Zheng et al. [2]. This strong alignment indicates that our MDLM reproduction is accurate. Since Loopholing is implemented directly on top of this verified reproduction, and also its training-time approximated PPL is at a comparable level, **it is not reasonable to claim that our implementation is incorrect based solely on a personal, subjective impression**.
> > > >
> > > >
> > > > [1] Sahoo, Subham Sekhar, et al. "The diffusion duality." arXiv preprint arXiv:2506.10892 (2025).
> > > >
> > > > [2] Zheng, Kaiwen, et al. "Masked diffusion models are secretly time-agnostic masked models and exploit inaccurate categorical sampling." arXiv preprint arXiv:2409.02908 (2024).
> > > >
> > > > [3] Sahoo, Subham, et al. "Simple and effective masked diffusion language models." Advances in Neural Information Processing Systems 37 (2024): 130136-130184.
> > > >
> > > > ## Lack of evaluation on meaningful downstream tasks. The benchmarks used (LM1B, OWT, small arithmetic puzzles) are extremely limited and do not demonstrate real improvements in generation quality.
> > > >
> > > > ### (1/2) The benchmarks are extremely limited.
> > > >
> > > > **The benchmarks we use are standard in discrete diffusion language model research,** as discussed in the reviewer’s previous comments and in our rebuttal to **[W1-b, W7, Q4]**.
> > > >
> > > >
> > > > The core papers on discrete diffusion language models, such as SEDD, MDLM, UDLM, DUO, and related models (see references in the our response for "Severe weaknesses in evaluation methodology"), almost all evaluate primarily on language modeling benchmarks like LM1B or OpenWebText and do not report the kind of challenging downstream suite the reviewer is requesting. If the current evaluation protocol were considered “extremely limited”, then essentially all prior work in this research direction would be subject to exactly the same criticism.
> > > >
> > > > Although we already cover the standard benchmarks with standard metrics, we agree that evaluating on a broader set of downstream tasks is generally desirable, but there are practical limits in an academic setting. Large scale evaluations on MMLU, GSM8K, summarization, long form generation, code, and many diverse benchmarks, as requested, require computational resources and engineering support that are typically not available to academic research groups.
> > > >
> > > > ### (2/2) Lack of evaluation on meaningful downstream tasks.
> > > >
> > > > **Our paper already includes downstream evaluations.** In Appendix D.2, we report results on Lambada as well as several additional benchmarks. Lambada is the only task among them where sample-generation quality is critical, and on this task we observe a substantial improvement over the baselines. The other datasets are evaluated in a likelihood-based setting, where generation quality has much less influence; therefore, large performance gains are not expected.
> > > >
> > > > While these downstream tasks are simpler than the industry-scale tasks the reviewer suggests, they are widely used and well-established benchmarks in academic-scale DDLM research.

---

> > > > > ### Author Response · Authors · 2025-11-28
> > > > >
> > > > > ## No comparison to strong SOTA autoregressive models
> > > > >
> > > > > **As mentioned earlier, our primary contribution is to demonstrate the potential of our proposed discrete diffusion models at the similar scale of the previous works, rather than to compete with SOTA autoregressive models.**
> > > > >
> > > > > Accordingly, our main evaluations focus on comparisons with major prior discrete diffusion models (MDLM, SEDD, UDLM) in the same experimental setting. For the **autoregressive baseline, we simply follow previous work and train a general Transformer based language model** with the same parameter size, data, and training budget, and use it as a reference point rather than a heavily tuned SOTA system.
> > > > >
> > > > > ## Lack of novelty relative to self-conditioning. Self-conditioning already exists in diffusion models.
> > > > >
> > > > > **We respectfully disagree.** Our primary contribution are the identification of the sampling wall in discrete diffusion and the introduction of **a new discrete diffusion model that incorporates both a stochastic token path and a deterministic prediction path**—a design that, to our knowledge, has not appeared in the existing Discrete Diffusion literature. A key challenge in such a model is how to train it without performing multistep backpropagation through the deterministic path. Recognizing this difficulty and showing that it can be effectively addressed by leveraging self-conditioning is itself a substantive technical contribution. Given this context, it is far too narrow to claim that our work is not novel merely because it employs self-conditioning.
> > > > >
> > > > > ## The paper does not provide ablations isolating the effect of the new component.
> > > > >
> > > > > **We would like to clarify that ablations the reviewer requested are indeed provided.** Among the four experiments the reviewer refers to, one is **already included in the original paper**, and the remaining three ablations were also **already provided in our rebuttal** with the following references: [W4-a, Q3-a], [W4-b, Q3-b,c], and [Q2]. The four experiments are as follows:
> > > > >
> > > > > 1.  isolating self-conditioning and latent propagation
> > > > > 2.  the latent path with single-pass training
> > > > > 3.  using $x_{\theta,t}$ instead of $h_t$
> > > > > 4.  comparison to standard self-conditioning (propagating sampled token instead of latent)
> > > > >
> > > > > ## Unsupported efficiency claims. The paper claims improved efficiency without reporting wall-clock speed, FLOPs, or token throughput.
> > > > >
> > > > > We would like to clarify that we do not claim our method is more efficient than prior work in absolute terms. Our statement refers specifically to the fact that the proposed model can be trained **more efficiently than a naïve implementation that backpropagates through the deterministic path**. We will make this distinction clearer in the revision. This is an algorithmic efficiency improvement—one that follows directly from avoiding multistep backpropagation—and is therefore evident even without measuring time or throughput. Nonetheless, we **already provided additional supporting results in the rebuttal** (**[W8, Q6]**).

---

### Meta-Review · Area_Chair_XJtS · 2026-01-05

**Summary:**

The paper introduces Loopholing Discrete Diffusion Models (LDDMs), which bypass the loss of information from discrete sampling in discrete diffusion by propagating a deterministic continuous latent pathway across denoising steps, while training this pathway efficiently via a self-conditioning scheme. Across masked/uniform diffusion backbones, LDDMs deliver large improvements in sample-based generation quality and perplexity on standard DDLM benchmarks, and improve accuracy on discrete arithmetic reasoning tasks.

Strengths
* Clear diagnosis of an important limitation in discrete diffusion (information collapse after categorical sampling) and a simple, implementable architectural fix (latent propagation).
* Strong empirical gains across multiple backbones and tasks, with analyses supporting the mechanism and addressing oscillation/idle-step hypotheses.
* Rebuttal strengthens experimental rigor: clearer novelty positioning vs self-conditioning, additional ablations, and additional sample-based evaluations/metrics.

Weaknesses
* Some novelty concerns relative to self-conditioning and prior discrete-sampling discussions (though the rebuttal clarifies the distinction and provides controlled comparisons).
* Evaluation remains largely within academic-scale DDLM protocols; broader downstream task coverage and stronger wall-clock/throughput reporting would further strengthen claims (with the rebuttal clarifying what “efficiency” means and adding supporting measurements).

Recommendation: Accept. The core idea is technically sound, the gains are substantial and well-supported, and the rebuttal materially addresses the main methodological and experimental concerns.

**Reviewer Concerns:**

The main concerns raised by dHL3, particularly around novelty vs. variants of self-conditioning, need for more detailed controlled comparisons/ablations, and non-PPL, sample-based evaluation, are valid points to raise. It is reassuring that the rebuttal directly addresses them via explicit ablations (including standard self-conditioning baselines), clearer novelty framing, and additional/clarified sample-quality metrics (e.g., LLM-as-judge, diversity, entropy, and discussion around Gen PPL degeneration). Remaining gaps are mostly about breadth (limited large-scale downstream tasks) and practical efficiency reporting beyond the clarified training-vs-inference story; these are reasonable limitations but do not undermine the main contribution.

**Reviewer Scores:**

* 2QKi (8) is unlikely to change materially (minor presentation clarifications only).
* Sjct (6) may move upward given the added scalability evidence and continued pre-training results plus clarification around UDLM via sample-based metrics.
* dHL3 (0/2) is the least likely to increase substantially given the stated stance, but to the extent the rebuttal addresses exactly their requested axes (novelty vs self-conditioning variants and non-PPL evaluations), a modest upward adjustment (if any) would be driven by those clarifications rather than the broader downstream-task request.

---

### Decision · Program_Chairs · 2026-01-26

Accept (Poster)